# State of the Art of Continuous and Atomistic Modeling of Electromechanical Properties of Semiconductor Quantum Dots

**DOI:** 10.3390/nano13121820

**Published:** 2023-06-07

**Authors:** Daniele Barettin

**Affiliations:** Daniele Barettin of Electronic Engineering, Università Niccoló Cusano, 00133 Rome, Italy; daniele.barettin@unicusano.it

**Keywords:** quantum dots, wurtzite, zincblende, electromechanical fields

## Abstract

The main intent of this paper is to present an exhaustive description of the most relevant mathematical models for the electromechanical properties of heterostructure quantum dots. Models are applied both to wurtzite and zincblende quantum dot due to the relevance they have shown for optoelectronic applications. In addition to a complete overview of the continuous and atomistic models for the electromechanical fields, analytical results will be presented for some relevant approximations, some of which are unpublished, such as models in cylindrical approximation or a cubic approximation for the transformation of a zincblende parametrization to a wurtzite one and vice versa. All analytical models will be supported by a wide range of numerical results, most of which are also compared with experimental measurements.

## 1. Introduction

Quantum dot heterostructures have recently received much attention due to their relevance for optoelectronic devices [1]. The electron spectrum of an ideal quantum dot is a set of discreet levels material-, shape- and size-dependent, with a density of states given by a delta function. Although an inhomogeneous broadening of spectra is usually caused by some size and shape distribution of quantum dots, the possibility of changing growth regimes leads to quantum dots with a different size, shape and density depending on the requirements. There are numerous potential applications for semiconductor quantum dots. To highlight a few recent examples, we can consider Polymer-QDs nanocomposites, which are extensively studied, and find utility in optical/electrical sensors, light-emitting diodes, as well as biological labeling/imaging [2]. Moreover, these nanocomposites hold promise as recyclable photocatalysts for aqueous PET-RAFT polymerization [3].

The investigation of electromechanical fields in nanostructures and their interplay is a subject that continues to attract growing interest. Semiconductor nanocrystals provide an excellent platform for studying the quantum interactions between conduction and valence electrons, phonons, and photons [4,5]. The ability to control various factors such as nanostructure geometry (size and shape), material properties, charge transport, impurities, and external fields offers the opportunity to tailor device characteristics for specific applications [5,6,7,8,9,10,11,12,13,14]. In order to optimize these processes for specific device applications, it is essential to determine the distribution of coupled strain and electric fields, which may involve piezoelectric and electrostrictive effects in some cases [15,16,17,18,19]. By obtaining the distributions of electromechanical fields, it becomes possible to evaluate properties related to quantum computing and optoelectronic devices, among others.

Traditionally, two approaches are employed to investigate electromechanical field effects: atomistic methods and continuum methods [20,21,22,23,24,25,26,27,28]. Atomistic methods are computationally intensive but offer detailed information at the inter-atomic scale, taking into account the complete crystal lattice structure and the specific positions of atoms associated with lattice points. These details become particularly significant as nanostructures approach dimensions comparable to the lattice constant. On the other hand, continuum models are computationally efficient and provide accurate results for nanostructures with dimensions significantly larger than the lattice constant. The advantage of continuum models lies in their ability to reduce computation time, making them preferable and often necessary for device calculations where multiphysics effects, such as electromechanical field interactions, play a crucial role.

The purpose of this study is to provide, in the first part, a mathematically comprehensive and exhaustive description of the two methods, considering important computational aspects such as boundary conditions and convenient approximations for model implementation. The second part presents a series of examples that, although they cannot cover the vast scientific production on the subject in recent decades, offer an overview of the various applications of the models to Zincblende, Wurtzite, and Polytypic quantum dots.

## 2. Fundamental Equations for Continuum Model

### 2.1. The Strain Tensor

Under the action of applied forces, solid bodies exhibits a deformation, changing their shape and volume [29]. If the position of any points of the body before the deformation is given by the positions vector $\vec{r}$ (with components $x_{1}=x$, $x_{2}=y$, $x_{3}=z$), after the deformation we have a new value for the position $\vec{r}{}^{\prime }$ (with components $x_{i}^{\prime }$). So we can define the following displacement vector:(1)$$\begin{array}{c} \vec{u}=\vec{r}{}^{\prime }-\vec{r}, \end{array}$$
with components $u_{i}=x_{i}^{\prime }-x_{i}$. The distance between two points of the body is given by $dl=\sqrt{dx_{1}^{2}+dx_{2}^{2}+dx_{3}^{2}}$ and $dl^{\prime }=\sqrt{dx_{1}^{{}^{\prime }2}+dx_{2}^{{}^{\prime }2}+dx_{3}^{{}^{\prime }2}}$ before and after the deformation, respectively. Using $dx_{i}^{\prime }=dx_{i}-du_{i}$ and substituting $du_{i}=\left(\partial u_{i}/\partial x_{k}\right)dx_{k}$ we can write using the general summation rule:(2)$$\begin{array}{c} dl^{{}^{\prime }2}=dl^{2}+2\frac{\partial u_{i}}{\partial x_{k}}dx_{i}dx_{k}+\frac{\partial u_{i}}{\partial x_{k}}\frac{\partial u_{i}}{\partial x_{l}}dx_{k}dx_{l}. \end{array}$$

Rearranging the second and third term we can finally write:(3)$$\begin{array}{c} dl^{{}^{\prime }2}=dl^{2}+2\varepsilon_{ik}dx_{i}dx_{k}, \end{array}$$
where we have defined a strain tensor:(4)$$\begin{array}{c} \varepsilon_{ik}=\frac{1}{2}\left(\frac{\partial u_{i}}{\partial x_{k}}+\frac{\partial u_{k}}{\partial x_{i}}+\frac{\partial u_{l}}{\partial x_{i}}\frac{\partial u_{l}}{\partial x_{k}}\right). \end{array}$$

In almost all general cases (for exceptions see Ref. [29]) if a body is subjected to a small deformation all the components of the strain tensor are small, so neglecting the last term in (Equation 4) as being of the second order of smallness we can write:(5)$$\begin{array}{c} \varepsilon_{ik}=\frac{1}{2}\left(\frac{\partial u_{i}}{\partial x_{k}}+\frac{\partial u_{k}}{\partial x_{i}}\right). \end{array}$$

### 2.2. The Stress Tensor

For a body in mechanical equilibrium the resultant of all the forces on every single portion of the body is equal to to zero. In THE case of deformation, some internal molecular forces, defined as internal stresses, tend to return the body to equilibrium. These forces, in THE absence of macroscopic electric fields, are near-action forces, which, acting on the considered portion of the body by the surrounding parts, effect only the surface of the portion itself.

If we want to express these forces, we have to consider the sum of all the forces on all the volume elements dV of that portion of bodies $\int \vec{f}dV$, where $\vec{f}$ is the force per unit volume. Because of Newton’s third law, they are equal to the sum of the forces exerted on the portion by the surrounding parts, i.d., from what we said above, equal to an integral over the surface of the portion.

From a theorem of vector analysis we know that an integral of a vector over an arbitrary volume can be transformed into an integral over the surface of the volume if the vector is the divergence of a tensor of rank two. This leads to:(6)$$\begin{array}{c} \int \vec{f_{i}}dV=\int \frac{\partial \sigma_{ik}}{\partial x_{k}}dV=\oint \sigma_{ik}dA_{k}, \end{array}$$
with
(7)$$\begin{array}{c} \vec{f_{i}}=\frac{\partial \sigma_{ik}}{\partial x_{k}}. \end{array}$$

The tensor $\sigma_{ik}$ is called the stress tensor, and $\sigma_{ik}dA_{k}$ is the *i*th component of the force on the surface elements $dA_{k}$. It is also possible to show that the stress tensor is symmetric [29]:(8)$$\begin{array}{c} \sigma_{ik}=\sigma_{ki}. \end{array}$$

At the mechanical equilibrium the internal stresses in every volume element must be balanced, so that ${\vec{f}}_{i}=0$. Thus the equilibrium equations for a deformed body can be written as:(9)$$\begin{array}{c} \frac{\partial \sigma_{ik}}{\partial x_{k}}=0. \end{array}$$

### 2.3. Free Energy

We want to derive now an expression of the free energy as a function of the strain tensor. Multiplying the force $\vec{f_{i}}=\frac{\partial \sigma_{ik}}{\partial x_{k}}$ by the displacement $\delta u_{i}$ and integrating over the volume *V* we can calculate the work δW done by the internal stress per unit volume:(10)$$\begin{array}{c} \int \delta WdV=\int \left(\frac{\partial \sigma_{ik}}{\partial x_{k}}\right)\delta u_{i}dV, \end{array}$$
integrating by parts and considering an infinite medium not deformed at infinity we have [29]:(11)$$\begin{array}{c} \delta W=\sigma_{ik}\delta \varepsilon_{ik}, \end{array}$$
so that we can write an expression for the internal energy dU at the thermodynamic equilibrium for a reversible process at temperature *T*:(12)$$\begin{array}{c} dU=TdS-dW=TdS-\sigma_{ik}d\varepsilon_{ik}, \end{array}$$
where *S* is the entropy of the system. Introducing the Helmholtz free energy of the body F=U−TS we obtain:(13)$$\begin{array}{c} dF=-SdT+\sigma_{ik}d\varepsilon_{ik}, \end{array}$$
which for constant temperature leads to:(14)$$\begin{array}{c} \sigma_{ik}=\left(\frac{\partial F}{\partial \varepsilon_{ik}}\right). \end{array}$$

The idea is to expand *F* in powers of $\varepsilon_{ik}$ with an assumption of small deformations, and, if we consider an isotropic deformed body at constant temperature, its undeformed state is a state in absence of any external forces, i.d., $\sigma_{ik}=0$, and because of (Equation 14) this implies that there are no linear terms in the expansion of *F*, which in terms of second order can be written as:(15)$$\begin{array}{c} F=F_{0}+\frac{1}{2}\lambda {\varepsilon_{ii}}^{2}+\mu {\varepsilon_{ik}}^{2}. \end{array}$$

The constant term $F_{0}$ is the free energy of the undeformed body, and we will omit it in the following. The quantities λ and μ are called *Lamè coefficients*. The expression of the free energy for a crystal in a compression at constant temperature is, such as for isotropic bodies, still a quadratic function of the strain tensor, but with a larger number of coefficients. The general form for a deformed crystal is given by [29]:(16)$$\begin{array}{c} F=\frac{1}{2}C_{iklm}\varepsilon_{ik}\varepsilon_{lm}, \end{array}$$
where $C_{iklm}$ is the rank four *elastic modulus tensor*, defined with the following symmetry properties:(17)$$\begin{array}{c} C_{iklm}=C_{kilm}=C_{ikml}=C_{lmik}. \end{array}$$

With (Equation 14) the stress tensor for a crystal in terms of strain tensor is given by:(18)$$\begin{array}{c} \sigma_{ik}=\left(\frac{\partial F}{\partial \varepsilon_{ik}}\right)=C_{iklm}\varepsilon_{lm}. \end{array}$$

The elastic modulus tensor is usually expressed also as $C_{\alpha \beta }$, with α and β taking values from 1 to 6 in correspondence with xx, yy, zz, yz, zx, xy.

### 2.4. Constitutive Relations

For a dielectric material there is an additional contribution given by AN electric field to Equation (Equation 12):(19)$$\begin{array}{c} dU^{elec.}=\vec{E}\cdot d\vec{D}=E_{i}dD_{i}, \end{array}$$
where $\vec{E}$ and $\vec{D}$ are the electric field and electric displacement vectors, respectively. So the total internal energy reads:(20)$$\begin{array}{c} dU=TdS+\sigma_{ik}d\varepsilon_{ik}+E_{i}dD_{i}, \end{array}$$
and it is also possible to define in a similar way another thermodynamic potential, the enthalpy:(21)$$\begin{array}{c} dH=TdS-\varepsilon_{ik}d\sigma_{ik}-D_{i}dE_{i}. \end{array}$$

For an isentropic process, by applying the chain rule to the derivates of *U* and *H*, we can express some thermodynamic identities:(22)$$\begin{array}{c} e_{ikl}=\left(\frac{\partial D_{i}}{\partial \varepsilon_{kl}}\right)=-\left(\frac{\partial \sigma_{kl}}{\partial E_{i}}\right)\;\left(\mathrm{piezoelectric}\;\mathrm{coefficient}\right) \end{array}$$
(23)$$\begin{array}{c} {\hat{\epsilon }}_{ik}=\left(\frac{\partial D_{i}}{\partial E_{k}}\right)\;\left(\mathrm{permittivity}\;\mathrm{tensor}\right) \end{array}$$
(24)$$\begin{array}{c} C_{iklm}=\left(\frac{\partial \sigma_{ik}}{\partial \varepsilon_{lm}}\right)\;(\mathrm{elastic}\;\mathrm{modulus}\;\mathrm{tensor}). \end{array}$$

For small isentropic variation we can therefore write:$$dD_{i}=\left(\frac{\partial D_{i}}{\partial E_{k}}\right)dE_{k}+\left(\frac{\partial D_{i}}{\partial \varepsilon_{kl}}\right)d\varepsilon_{kl}={\hat{\epsilon }}_{ik}dE_{k}+e_{ikl}d\varepsilon_{kl}$$
(25)$$\begin{array}{c} d\sigma_{kl}=\left(\frac{\partial \sigma_{kl}}{\partial E_{i}}\right)dE_{i}+\left(\frac{\partial \sigma_{kl}}{\partial \varepsilon_{mn}}\right)d\varepsilon_{mn}=-e_{ikl}dE_{i}+C_{klmn}d\varepsilon_{mn}, \end{array}$$
so we can write the constitutive relations for the relative finite quantities:$$D_{i}={\hat{\epsilon }}_{ik}E_{k}+e_{ikl}\varepsilon_{kl}$$
(26)$$\begin{array}{c} \sigma_{kl}=-e_{ikl}E_{i}+C_{klmn}\varepsilon_{mn}, \end{array}$$

### 2.5. Strain Field in Quantum Dot

The lattice constants in semiconductor heterostructures vary with coordinates, and the lattice mismatch between the quantum dot structure and the matrix material in which it is embedded generates an intrinsic local strain field different from zero [25]. The free elastic energy can be written as:(27)$$\begin{array}{c} F=\int_{V}d\vec{r}\frac{1}{2}C_{iklm}\left(\vec{r}\right)\varepsilon_{ik}\left(\vec{r}\right)\varepsilon_{lm}\left(\vec{r}\right), \end{array}$$
where *V* is the total volume of the system. To take into account lattice mismatch (cfr. Figure 1), the strain tensor is represented as (i,j=x,y,z):(28)$$\begin{array}{c} \varepsilon_{ij}=\varepsilon_{ij}^{\left(u\right)}+\varepsilon_{ij}^{\left(0\right)}, \end{array}$$
where $\varepsilon_{ij}^{\left(0\right)}$ is the tensor of local intrinsic strain and $\varepsilon_{ij}^{\left(u\right)}$ is the local strain tensor dependent on positions [30], given by (Equation 5). The contribution of internal strain has been neglected in Equation (Equation 28).

#### 2.5.1. Zincblende Quantum Dot

The elastic energy density for a crystal with zincblende symmetry read [25]:(29)$$\begin{array}{c} \;F=\frac{1}{2}\left[C_{11}\left(\varepsilon_{xx}^{2}+\varepsilon_{yy}^{2}+\varepsilon_{zz}^{2}\right)+2C_{12}\left(\varepsilon_{xx}\varepsilon_{yy}+\varepsilon_{xx}\varepsilon_{zz}+\varepsilon_{yy}\varepsilon_{zz}\right)+4C_{44}\left(\varepsilon_{xy}^{2}+\varepsilon_{xz}^{2}+\varepsilon_{yz}^{2}\right)\right], \end{array}$$
since the only linearly independent elastic constants for a zincblende structure are given by:(30)$$\begin{array}{c} C_{1111}\equiv C_{11}\;C_{1122}\equiv C_{12}\;C_{2323}\equiv C_{44}. \end{array}$$

The intrinsic strain tensor is given by:(31)$$\begin{array}{c} \varepsilon_{ij}^{\left(0\right)}=\delta_{ij}a, \end{array}$$
with $a=\frac{a_{matrix}-a_{QD}}{a_{matrix}}$ in the dot and zero otherwise. Here $a_{matrix}$ and $a_{QD}$ are the lattice constants of the matrix and the quantum dot, respectively.

#### 2.5.2. Wurtzite Quantum Dot

The elastic energy density for a crystal with wurtzite symmetry is given by [25]:(32)$$\begin{array}{c} F=\frac{1}{2}\left[C_{11}\left(\varepsilon_{xx}^{2}+\varepsilon_{yy}^{2}\right)+C_{33}\varepsilon_{zz}^{2}+2C_{12}\varepsilon_{xx}\varepsilon_{yy}+2C_{13}\varepsilon_{zz}\left(\varepsilon_{xx}+\varepsilon_{yy}\right)+4C_{44}\left(\varepsilon_{xz}^{2}+\varepsilon_{yz}^{2}\right)+2\left(C_{11}-C_{12}\right)\varepsilon_{xy}^{2}\right] \end{array}$$

In a crystal with wurtzite symmetry the linearly independent elastic constants are:(33)$$\begin{array}{cc} C_{1111} & \equiv C_{11};\;C_{1122}\equiv C_{12};\;C_{1133}\equiv C_{13};\;\\ C_{3333} & \equiv C_{33};\;C_{2323}\equiv C_{44};\;C_{2121}\equiv \left(C_{11}-C_{12}\right)/2. \end{array}$$

The tensor of local intrinsic strain is given by:(34)$$\begin{array}{c} \varepsilon_{ij}^{\left(0\right)}=\left(\delta_{ij}-\delta_{iz}\delta_{jz}\right)a+\delta_{iz}\delta_{jz}c, \end{array}$$
with $a=\frac{a_{matrix}-a_{QD}}{a_{matrix}}$ and $c=\frac{c_{matrix}-c_{QD}}{c_{matrix}}$ the lattice mismatch in the hexagonal plane and the *c* axis, respectively. The parameters $a_{matrix}$, $c_{matrix}$, and $a_{QD}$, $c_{QD}$ are the lattice constants of the matrix and the quantum dot, respectively. The *c* axis is the axis of the sixfold rotational symmetry of the wurtzite material, which we take coincident with the *z* axis.

### 2.6. Piezoelectric Field in Quantum Dot

Under an applied stress some semiconductors develop an electric moment with magnitude proportional to the stress [25,31]. The induced polarization is related to the strain tensor by the piezoelectric coefficients (Equation 22):(35)$$\begin{array}{c} P_{i}\left(\vec{r}\right)=e_{ilm}\left(\vec{r}\right)\varepsilon_{lm}\left(\vec{r}\right), \end{array}$$
the index ilm run over the spatial coordinates, namely:(36)$$\begin{array}{c} \left(\begin{array}{cccccc} e_{111} & e_{122} & e_{133} & e_{123} & e_{113} & e_{112}\\ e_{211} & e_{222} & e_{233} & e_{223} & e_{213} & e_{212}\\ e_{311} & e_{322} & e_{333} & e_{323} & e_{313} & e_{312} \end{array}\right). \end{array}$$

For some wurtzite nitrides we have also to take in account a spontaneous polarization $P_{sp}$, whose polarity depends on the last anion or cation at the surface. The total polarization generates a piezoelectric field $E_{P}$, which in the absence of external charges can be evaluated by solving the Maxwell-Poisson equation:(37)$$\begin{array}{c} \vec{\nabla }\cdot \vec{D}\left(\vec{r}\right)=0, \end{array}$$
where the displacement vector $\vec{D}$ is given by Equation (Equation 26).

#### 2.6.1. Zincblende Quantum Dot

Converting from tensor notation to matrix notation by:(38)$$\begin{array}{c} e_{ilm}=\left\{\begin{array}{cc} e_{ik}, & k=1,2,3;\\ \frac{1}{2}e_{ik}, & k=4,5,6; \end{array}\right., \end{array}$$
the independent piezoelectric coefficients for a zincblende structure are given by:(39)$$\begin{array}{c} \left(\begin{array}{cccccc} 0 & 0 & 0 & e_{14} & 0 & 0\\ 0 & 0 & 0 & 0 & e_{14} & 0\\ 0 & 0 & 0 & 0 & 0 & e_{14} \end{array}\right), \end{array}$$
so that we can write for the polarization in terms of components:(40)$$\begin{array}{c} \left.\begin{array}{c} P_{x}=e_{14}\varepsilon_{yz},\\ P_{y}=e_{14}\varepsilon_{xz},\\ P_{z}=e_{14}\varepsilon_{xy}, \end{array}\right. \end{array}$$
while for the permittivity tensor we have:(41)$$\begin{array}{c} {\hat{\epsilon }}_{Zb}=\left(\begin{array}{ccc} \hat{\epsilon } & 0 & 0\\ 0 & \hat{\epsilon } & 0\\ 0 & 0 & \hat{\epsilon } \end{array}\right), \end{array}$$
where $\hat{\epsilon }$ is a constant value material dependent.

#### 2.6.2. Wurtzite Quantum Dot

In a crystal with wurtzite symmetry the only nonzero component for the spontaneous polarization is along the *z*-axis (the axis of the sixfold rotational symmetry, as we already mentioned), while the independent piezoelectric coefficients are:(42)$$\begin{array}{c} \left(\begin{array}{cccccc} 0 & 0 & 0 & 0 & e_{15} & 0\\ 0 & 0 & 0 & e_{15} & 0 & 0\\ e_{31} & e_{31} & e_{33} & 0 & 0 & 0 \end{array}\right), \end{array}$$
so nominally the components of the polarization are:(43)$$\begin{array}{c} \left.\begin{array}{c} P_{x}=e_{15}\varepsilon_{xz},\\ P_{y}=e_{15}\varepsilon_{yz},\\ P_{z}=e_{31}\left(\varepsilon_{xx}+\varepsilon_{yy}\right)+e_{33}\varepsilon_{zz}+P_{sp}, \end{array}\right. \end{array}$$
and the permittivity tensor is given by:(44)$$\begin{array}{c} {\hat{\epsilon }}_{Wz}=\left(\begin{array}{ccc} {\hat{\epsilon }}_{11} & 0 & 0\\ 0 & {\hat{\epsilon }}_{11} & 0\\ 0 & 0 & {\hat{\epsilon }}_{33} \end{array}\right), \end{array}$$
with ${\hat{\epsilon }}_{11}$ and ${\hat{\epsilon }}_{33}$ constant values material dependent.

### 2.7. Governing Equations for Electromechanical Fields

The governing equations for the electromechanical fields in an heterostructures quantum dot are given by the already mentioned equilibrium Equation (Equation 9)— also known as Navier’s static equation–and Maxwell-Poisson Equation (Equation 37) which we rewrite here:$$\frac{\partial \sigma_{ij}}{\partial x_{j}}=0$$
(45)$$\begin{array}{c} \;\nabla \cdot D=0, \end{array}$$

From Equation (Equation 45) we obtain a set of four coupled equations in the electromechanical fields. The expression for the stress tensor and the electric displacement are given by the constitutive relations (Equation 26), which we express in a more convenient way here:(46)$$\begin{array}{c} \sigma_{ik}=C_{iklm}\varepsilon_{lm}+e_{ikn}\frac{\partial V}{\partial x_{i}},\;D_{i}=-\varepsilon_{in}\frac{\partial V}{\partial x_{i}}+e_{ilm}\varepsilon_{lm}+P_{sp,i}, \end{array}$$
where *V* is the electric potential.

#### 2.7.1. Zincblende Quantum Dot

Equation (Equation 45) using (Equation 5) for a zincblende structure read nominally:(47)$$\begin{array}{cc} \frac{\partial }{\partial x} & \left(C_{11}\frac{\partial u_{x}}{\partial x}\right)+\frac{\partial }{\partial y}\left(C_{44}\frac{\partial u_{x}}{\partial y}\right)+\frac{\partial }{\partial z}\left(C_{44}\frac{\partial u_{x}}{\partial z}\right)+\frac{\partial }{\partial x}\left(C_{12}\frac{\partial u_{y}}{\partial y}\right)+\frac{\partial }{\partial y}\left(C_{44}\frac{\partial u_{y}}{\partial x}\right)\\ + & \frac{\partial }{\partial x}\left(C_{12}\frac{\partial u_{z}}{\partial z}\right)+\frac{\partial }{\partial z}\left(C_{44}\frac{\partial u_{z}}{\partial x}\right)+\frac{\partial }{\partial y}\left(e_{14}\frac{\partial V}{\partial z}\right)+\frac{\partial }{\partial z}\left(e_{14}\frac{\partial V}{\partial y}\right)+\frac{\partial }{\partial x}[(C_{11}+2C_{12})a]=0\\ \frac{\partial }{\partial x} & \left(C_{44}\frac{\partial u_{x}}{\partial y}\right)+\frac{\partial }{\partial y}\left(C_{12}\frac{\partial u_{x}}{\partial x}\right)+\frac{\partial }{\partial x}\left(C_{44}\frac{\partial u_{y}}{\partial x}\right)+\frac{\partial }{\partial y}\left(C_{11}\frac{\partial u_{y}}{\partial y}\right)+\frac{\partial }{\partial z}\left(C_{44}\frac{\partial u_{y}}{\partial z}\right)\\ + & \frac{\partial }{\partial y}\left(C_{12}\frac{\partial u_{z}}{\partial z}\right)+\frac{\partial }{\partial z}\left(C_{44}\frac{\partial u_{z}}{\partial y}\right)+\frac{\partial }{\partial x}\left(e_{14}\frac{\partial V}{\partial z}\right)+\frac{\partial }{\partial z}\left(e_{14}\frac{\partial V}{\partial x}\right)+\frac{\partial }{\partial y}[(C_{11}+2C_{12})a]=0\\ \frac{\partial }{\partial x} & \left(C_{44}\frac{\partial u_{x}}{\partial z}\right)+\frac{\partial }{\partial z}\left(C_{12}\frac{\partial u_{x}}{\partial x}\right)+\frac{\partial }{\partial y}\left(C_{44}\frac{\partial u_{y}}{\partial z}\right)+\frac{\partial }{\partial z}\left(C_{12}\frac{\partial u_{y}}{\partial y}\right)+\frac{\partial }{\partial x}\left(C_{44}\frac{\partial u_{z}}{\partial x}\right)\\ + & \frac{\partial }{\partial y}\left(C_{44}\frac{\partial u_{z}}{\partial y}\right)+\frac{\partial }{\partial z}\left(C_{11}\frac{\partial u_{z}}{\partial z}\right)+\frac{\partial }{\partial x}\left(e_{14}\frac{\partial V}{\partial y}\right)+\frac{\partial }{\partial y}\left(e_{14}\frac{\partial V}{\partial x}\right)+\frac{\partial }{\partial z}[(C_{11}+2C_{12})a]=0\\ \frac{\partial }{\partial y} & \left(e_{14}\frac{\partial u_{x}}{\partial z}\right)+\frac{\partial }{\partial z}\left(e_{14}\frac{\partial u_{x}}{\partial y}\right)+\frac{\partial }{\partial x}\left(e_{14}\frac{\partial u_{y}}{\partial z}\right)+\frac{\partial }{\partial z}\left(e_{14}\frac{\partial u_{y}}{\partial x}\right)+\frac{\partial }{\partial x}\left(e_{14}\frac{\partial u_{z}}{\partial y}\right)\\ + & \frac{\partial }{\partial y}\left(e_{14}\frac{\partial u_{z}}{\partial x}\right)+\frac{\partial }{\partial x}\left(\hat{\varepsilon }\frac{\partial V}{\partial x}\right)+\frac{\partial }{\partial y}\left(\hat{\varepsilon }\frac{\partial V}{\partial y}\right)+\frac{\partial }{\partial z}\left(\hat{\varepsilon }\frac{\partial V}{\partial z}\right)=0 \end{array}$$

#### 2.7.2. Wurtzite Quantum Dot

The explicit expression for Equation (Equation 45) using (Equation 5) is given by:(48)$$\begin{array}{cc} \frac{\partial }{\partial x} & \left(C_{11}\frac{\partial u_{x}}{\partial x}\right)+\frac{\partial }{\partial y}\left[\frac{\left(C_{11}-C_{12}\right)}{2}\frac{\partial u_{x}}{\partial y}\right]+\frac{\partial }{\partial z}\left(C_{44}\frac{\partial u_{x}}{\partial z}\right)+\frac{\partial }{\partial x}\left(C_{12}\frac{\partial u_{y}}{\partial y}\right)\\ + & \frac{\partial }{\partial y}\left[\frac{\left(C_{11}-C_{12}\right)}{2}\frac{\partial u_{y}}{\partial x}\right]+\frac{\partial }{\partial x}\left(C_{13}\frac{\partial u_{z}}{\partial z}\right)+\frac{\partial }{\partial z}\left(C_{44}\frac{\partial u_{z}}{\partial x}\right)\\ + & \frac{\partial }{\partial x}\left(e_{31}\frac{\partial V}{\partial z}\right)+\frac{\partial }{\partial z}\left(e_{15}\frac{\partial V}{\partial x}\right)+\frac{\partial }{\partial x}[(C_{11}+C_{12})a+C_{13}c]=0\\ \frac{\partial }{\partial x} & \left[\frac{\left(C_{11}-C_{12}\right)}{2}\frac{\partial u_{x}}{\partial y}\right]+\frac{\partial }{\partial y}\left(C_{12}\frac{\partial u_{x}}{\partial x}\right)+\frac{\partial }{\partial x}\left[\frac{\left(C_{11}-C_{12}\right)}{2}\frac{\partial u_{y}}{\partial x}\right]\\ + & \frac{\partial }{\partial y}\left(C_{11}\frac{\partial u_{y}}{\partial y}\right)+\frac{\partial }{\partial z}\left(C_{44}\frac{\partial u_{y}}{\partial z}\right)+\frac{\partial }{\partial y}\left(C_{13}\frac{\partial u_{z}}{\partial z}\right)+\frac{\partial }{\partial z}\left(C_{44}\frac{\partial u_{z}}{\partial y}\right)\\ + & \frac{\partial }{\partial y}\left(e_{31}\frac{\partial V}{\partial z}\right)+\frac{\partial }{\partial z}\left(e_{15}\frac{\partial V}{\partial y}\right)+\frac{\partial }{\partial y}[(C_{11}+C_{12})a+C_{13}c]=0\\ \frac{\partial }{\partial x} & \left(C_{44}\frac{\partial u_{x}}{\partial z}\right)+\frac{\partial }{\partial z}\left(C_{13}\frac{\partial u_{x}}{\partial x}\right)+\frac{\partial }{\partial y}\left(C_{44}\frac{\partial u_{y}}{\partial z}\right)+\frac{\partial }{\partial z}\left(C_{13}\frac{\partial u_{y}}{\partial y}\right)\\ + & \frac{\partial }{\partial x}\left(C_{44}\frac{\partial u_{z}}{\partial x}\right)+\frac{\partial }{\partial y}\left(C_{44}\frac{\partial u_{z}}{\partial y}\right)+\frac{\partial }{\partial z}\left(C_{33}\frac{\partial u_{z}}{\partial z}\right)+\frac{\partial }{\partial x}\left(e_{15}\frac{\partial V}{\partial x}\right)\\ + & \frac{\partial }{\partial y}\left(e_{15}\frac{\partial V}{\partial y}\right)+\frac{\partial }{\partial z}\left(e_{33}\frac{\partial V}{\partial z}\right)\frac{\partial }{\partial z}[2C_{13}a+C_{33}c]=0\\ \frac{\partial }{\partial x} & \left(e_{15}\frac{\partial u_{x}}{\partial z}\right)+\frac{\partial }{\partial z}\left(e_{31}\frac{\partial u_{x}}{\partial x}\right)+\frac{\partial }{\partial y}\left(e_{15}\frac{\partial u_{y}}{\partial z}\right)+\frac{\partial }{\partial z}\left(e_{31}\frac{\partial u_{y}}{\partial y}\right)\\ + & \frac{\partial }{\partial x}\left(e_{15}\frac{\partial u_{z}}{\partial x}\right)+\frac{\partial }{\partial y}\left(e_{15}\frac{\partial u_{z}}{\partial y}\right)+\frac{\partial }{\partial z}\left(e_{33}\frac{\partial u_{z}}{\partial z}\right)+\frac{\partial }{\partial x}\left({\hat{\varepsilon }}_{11}\frac{\partial V}{\partial x}\right)\\ + & \frac{\partial }{\partial y}\left({\hat{\varepsilon }}_{11}\frac{\partial V}{\partial y}\right)+\frac{\partial }{\partial z}\left({\hat{\varepsilon }}_{33}\frac{\partial V}{\partial z}\right)+\frac{\partial P_{sp}}{\partial z}=0 \end{array}$$

### 2.8. Electromechanical Fields in Cylindrical Coordinates

In some configurations, it is possible to solve the electromechanical problem using a cylindrical approximation. For a more detailed discussion on different aspects related to the cylindrical approximation, it is possible refer to Refs. [32,33,34,35,36,37]. We want to derive an expression for the strain tensors and the free elastic energy in cylindrical coordinates:(49)$$\begin{array}{c} \left\{\begin{array}{c} x=\varrho cos\varphi \\ y=\varrho sin\varphi \\ z=z \end{array},\right. \end{array}$$
so first we give the general transformation rules for a tensor *w* in a new system of coordinates [38]: (50)$$\begin{array}{cc} w_{ij}^{\prime } & =\alpha_{ik}\alpha_{jm}w_{km}\;\left(\mathrm{Second}\;\mathrm{rank}\;\mathrm{tensor}\right), \end{array}$$(51)$$\begin{array}{cc} w_{ijk}^{\prime } & =\alpha_{ir}\alpha_{js}\alpha_{kt}w_{rst}\;\left(\mathrm{Third}\;\mathrm{rank}\;\mathrm{tensor}\right), \end{array}$$(52)$$\begin{array}{cc} w_{ijkl}^{\prime } & =\alpha_{im}\alpha_{jn}\alpha_{kp}\alpha_{lq}w_{mnpq}\;\left(\mathrm{Fourth}\;\mathrm{rank}\;\mathrm{tensor}\right), \end{array}$$
where α are the direction cosines of the axes of the new system with respect to the old one. Using these rules we derive the following relations between the strain tensor and the displacement vectors in a system of cylindrical coordinates:(53)$$\begin{array}{cc} \varepsilon_{\varrho \varrho } & =\frac{\partial u_{\varrho }}{\partial \varrho },\;\varepsilon_{\varphi \varphi }=\frac{1}{\varrho }\frac{\partial u_{\varphi }}{\partial \varphi }+\frac{u_{\varrho }}{\varrho },\;\varepsilon_{zz}=\frac{\partial u_{z}}{\partial z},\\ \varepsilon_{\varrho \varphi } & =\frac{1}{2}\left(\frac{1}{\varrho }\frac{\partial u_{\varrho }}{\partial \varphi }+\frac{\partial u_{\varphi }}{\partial \varrho }-\frac{u_{\varphi }}{\varrho }\right),\;\varepsilon_{\varrho z}=\frac{1}{2}\left(\frac{\partial u_{\varrho }}{\partial z}+\frac{\partial u_{z}}{\partial \varrho }\right),\;\varepsilon_{\varphi z}=\frac{1}{2}\left(\frac{\partial u_{\varphi }}{\partial z}+\frac{1}{\varrho }\frac{\partial u_{z}}{\partial \varphi }\right), \end{array}$$
and we can also derive some useful identities between the strain tensor in cartesian and cylindrical coordinates:(54)$$\begin{array}{c} \varepsilon_{xx}+\varepsilon_{yy}=\varepsilon_{\varrho \varrho }+\varepsilon_{\varphi \varphi },\\ \varepsilon_{zz}=\varepsilon_{zz},\\ {\varepsilon_{xz}}^{2}+{\varepsilon_{yz}}^{2}={\varepsilon_{\varrho z}}^{2}+{\varepsilon_{\varphi z}}^{2},\\ {\varepsilon_{xx}}^{2}+{\varepsilon_{yy}}^{2}+2{\varepsilon_{xy}}^{2}={\varepsilon_{\varrho \varrho }}^{2}+{\varepsilon_{\varphi \varphi }}^{2}+2{\varepsilon_{\varrho \varphi }}^{2},\\ \varepsilon_{xx}\varepsilon_{yy}-{\varepsilon_{xy}}^{2}=\varepsilon_{\varrho \varrho }\varepsilon_{\varphi \varphi }-{\varepsilon_{\varrho \varphi }}^{2}, \end{array}$$
which can be used to write an expression for the free elastic energy both for a zincblende structure:(55)$$\begin{array}{c} F=\frac{1}{2}C_{11}\left(\varepsilon_{\varrho \varrho }^{2}+\varepsilon_{\varphi \varphi }^{2}+\varepsilon_{zz}^{2}\right)+C_{12}\left(\varepsilon_{\varrho \varrho }\varepsilon_{\varphi \varphi }+\varepsilon_{\varphi \varphi }\varepsilon_{zz}+\varepsilon_{\varrho \varrho }\varepsilon_{zz}\right)+2C_{44}\left(\varepsilon_{\varrho \varphi }^{2}+\varepsilon_{\varphi z}^{2}+\varepsilon_{\varrho z}^{2}\right), \end{array}$$
and for a wurtzite structure:(56)$$\begin{array}{cc} F & =\frac{1}{2}C_{11}\left(\varepsilon_{\varrho \varrho }^{2}+\varepsilon_{\varphi \varphi }^{2}\right)+\frac{1}{2}C_{33}\varepsilon_{zz}^{2}+C_{12}\varepsilon_{\varrho \varrho }\varepsilon_{\varphi \varphi }+C_{13}\left(\varepsilon_{\varphi \varphi }\varepsilon_{zz}+\varepsilon_{\varrho \varrho }\varepsilon_{zz}\right)\\ \; & +2C_{44}\left(\varepsilon_{\varphi z}^{2}+\varepsilon_{\varrho z}^{2}\right)+\left(C_{11}-C_{12}\right)\varepsilon_{\varrho \varphi }^{2}. \end{array}$$

#### 2.8.1. Governing Equations in Cylindrical Coordinates for a Zincblende Quantum Dot

Zincblende materials are not axisymmetric, so in order to reduce the problem to a two dimensional model in cylindrical coordinates, we need to make an isotropic assumptions. This entails that we disregard piezoelectricity in this model. To derive this assumption, we write the elastic tensors in a new system of complex coordinates:(57)$$\begin{array}{c} \left\{\begin{array}{c} \xi =x+iy\\ \eta =x-iy \end{array}\right., \end{array}$$

Because of (Equation 17) we write in this system:(58)$$\begin{array}{c} C_{\xi \eta \xi \eta }=C_{\eta \xi \xi \eta }=C_{\xi \eta \eta \xi }=C_{\eta \xi \eta \xi }, \end{array}$$
and
(59)$$\begin{array}{c} C_{\xi \xi \eta \eta }=C_{\eta \eta \xi \xi }. \end{array}$$

Using the properties of transformation (Equation 52) for a fourth order tensor we get:(60)$$\begin{array}{c} \left\{\begin{array}{c} C_{xxxx}=2C_{\xi \xi \xi \xi }+4C_{\xi \eta \xi \eta }+2C_{\xi \xi \eta \eta }\\ C_{xxyy}=-2C_{\xi \xi \xi \xi }+4C_{\xi \eta \xi \eta }-2C_{\xi \xi \eta \eta }\\ C_{xyxy}=-2C_{\xi \xi \xi \xi }+2C_{\xi \xi \eta \eta } \end{array}\right.. \end{array}$$

To impose an isotropic cylindrical symmetry we apply a rotation to the coordinates:(61)$$\begin{array}{c} \left\{\begin{array}{c} \xi^{\prime }=\xi e^{i\varphi }\\ \eta^{\prime }=\eta e^{-i\varphi } \end{array}\right., \end{array}$$
with the constrain that the elastic tensor must be independent on this rotation, which leads to:(62)$$\begin{array}{c} \left\{\begin{array}{c} C_{\xi^{\prime }\xi^{\prime }\xi^{\prime }\xi^{\prime }}=e^{i4\varphi }C_{\xi \xi \xi \xi }\\ C_{\xi^{\prime }\eta^{\prime }\xi^{\prime }\eta^{\prime }}=C_{\xi \eta \xi \eta }\\ C_{\xi^{\prime }\xi^{\prime }\eta^{\prime }\eta^{\prime }}=C_{\xi \xi \eta \eta } \end{array}\right.. \end{array}$$

The constraint is satisfied only if $C_{\xi \xi \xi \xi }=0$, so Equation (Equation 60) becomes:(63)$$\begin{array}{c} \left\{\begin{array}{c} C_{xxxx}=4C_{\xi \eta \xi \eta }+2C_{\xi \xi \eta \eta }\\ C_{xxyy}=4C_{\xi \eta \xi \eta }-2C_{\xi \xi \eta \eta }\\ C_{xyxy}=2C_{\xi \xi \eta \eta } \end{array}\right., \end{array}$$
which gives:(64)$$\begin{array}{c} C_{xxxx}-C_{xxyy}=2C_{xyxy}, \end{array}$$
which using the compact notation for the elastic tensors can be written as:(65)$$\begin{array}{c} 2C_{44}+C_{12}=C_{11}, \end{array}$$
which is the cylindrical isotropic assumption we were looking for. By applying this assumption we can derive the constitutive relations for a zincblende quantum dot in cylindrical coordinates:(66)$$\begin{array}{cc} \sigma_{\varrho \varrho } & =C_{12}\left(\varepsilon_{\varrho \varrho }+\varepsilon_{\varphi \varphi }+\varepsilon_{zz}+3a\right)+2C_{44}\left(\varepsilon_{\varrho \varrho }+a\right)\\ \sigma_{\varphi \varphi } & =C_{12}\left(\varepsilon_{\varrho \varrho }+\varepsilon_{\varphi \varphi }+\varepsilon_{zz}+3a\right)+2C_{44}\left(\varepsilon_{\varphi \varphi }+a\right)\\ \sigma_{zz} & =C_{12}\left(\varepsilon_{\varrho \varrho }+\varepsilon_{\varphi \varphi }+\varepsilon_{zz}+3a\right)+2C_{44}\left(\varepsilon_{zz}+a\right)\\ \sigma_{\varrho \varphi } & =4C_{44}\varepsilon_{\varrho \varphi }\\ \sigma_{\varphi z} & =4C_{44}\varepsilon_{\varphi z}\\ \sigma_{\varrho z} & =4C_{44}\varepsilon_{\varrho z}, \end{array}$$

Using the isotropic assumption (Equation 65) we can derive a set of equations governing the rotational invariant in cylindrical coordinates for a zincblende quantum dot for the strain fields, and to separate the problem in a (ϱ,z) part and a φ part. With adequate boundary conditions we can remove the angular dependence, which in terms of displacement leads to $u_{\varphi }=0$; therefore we write here only the equations ϱ- and z-dependent, which we have used in our two-dimensional models:(67)$$\begin{array}{cc} \frac{\partial }{\partial \varrho } & \left[\left(C_{12}+2C_{44}\right)\frac{\partial u_{\varrho }}{\partial \varrho }\right]+\frac{\partial }{\partial z}\left(C_{44}\frac{\partial u_{z}}{\partial z}\right)+\frac{\partial }{\partial \varrho }\left(C_{12}\frac{\partial u_{z}}{\partial z}\right)+\frac{\partial }{\partial z}\left(C_{44}\frac{\partial u_{z}}{\partial \varrho }\right)\\ + & \frac{\partial }{\partial \varrho }\left(\frac{C_{12}}{\varrho }u_{\varrho }\right)-\frac{2C_{44}}{\varrho^{2}}u_{\varrho }+\frac{2C_{44}}{\varrho }\frac{\partial u_{\varrho }}{\partial \varrho }+\frac{\partial }{\partial \varrho }[(3C_{12}+2C_{44})a]=0\\ \frac{\partial }{\partial \varrho } & \left(C_{44}\frac{\partial u_{\varrho }}{\partial z}\right)+\frac{\partial }{\partial z}\left(C_{12}\frac{\partial u_{\varrho }}{\partial \varrho }\right)+\frac{\partial }{\partial \varrho }\left(C_{44}\frac{\partial u_{z}}{\partial \varrho }\right)+\frac{\partial }{\partial z}\left[\left(C_{12}+2C_{44}\right)\frac{\partial u_{z}}{\partial z}\right]\\ + & \frac{\partial }{\partial z}\left(\frac{C_{12}}{\varrho }u_{\varrho }\right)+\frac{C_{44}}{\varrho }\frac{\partial u_{\varrho }}{\partial z}++\frac{C_{44}}{\varrho }\frac{\partial u_{z}}{\partial \varrho }+\frac{\partial }{\partial z}[(3C_{12}+2C_{44})a]=0. \end{array}$$

#### 2.8.2. Governing Equations in Cylindrical Coordinates for a Wurtzite Quantum Dot

In cylindrical coordinates the constitutive relations (Equation 46) for a wurtzite quantum dot take the form:(68)$$\begin{array}{cc} \sigma_{\varrho \varrho } & =C_{11}\varepsilon_{\varrho \varrho }+C_{12}\varepsilon_{\varphi \varphi }+C_{13}\varepsilon_{zz}+\left(C_{11}+C_{12}\right)a+C_{33}c+e_{31}\frac{\partial V}{\partial z},\\ \sigma_{\varphi \varphi } & =C_{11}\varepsilon_{\varphi \varphi }+C_{12}\varepsilon_{\varrho \varrho }+C_{13}\varepsilon_{zz}+\left(C_{11}+C_{12}\right)a+C_{33}c+e_{31}\frac{\partial V}{\partial z},\\ \sigma_{zz} & =C_{33}\varepsilon_{zz}+C_{13}\left(\varepsilon_{\varrho \varrho }+\varepsilon_{\varphi \varphi }\right)+2C_{13}a+C_{33}c+e_{33}\frac{\partial V}{\partial z},\\ \sigma_{\varrho \varphi } & =\left(C_{11}-C_{12}\right)\varepsilon_{r\varphi },\\ \sigma_{\varphi z} & =2C_{44}\varepsilon_{\varphi z}+e_{15}\left(\frac{1}{\varrho }\right)\frac{\partial V}{\partial \varphi },\\ \sigma_{\varrho z} & =2C_{44}\varepsilon_{\varrho z}+e_{15}\frac{\partial V}{\partial \varrho },\\ D_{\varrho } & =2e_{15}\varepsilon_{\varrho z}-{\hat{\varepsilon }}_{11}\frac{\partial V}{\partial \varrho },\\ D_{\varphi } & =2e_{15}\varepsilon_{\varphi z}-{\hat{\varepsilon }}_{11}\left(\frac{1}{\varrho }\right)\frac{\partial V}{\partial \varphi },\\ D_{z} & =e_{31}\left(\varepsilon_{\varrho \varrho }+\varepsilon_{\varphi \varphi }\right)+e_{33}\varepsilon_{zz}-{\hat{\varepsilon }}_{33}\frac{\partial V}{\partial z}+P_{sp}, \end{array}$$
and inserting these relations in Equation (Equation 45) we obtain a set of coupled equilibrium equations for the strain and the electrical fields in cylindrical coordinates:(69)$$\begin{array}{cc} \frac{\partial \sigma_{\varrho \varrho }}{\partial \varrho } & +\left(\frac{1}{\varrho }\right)\frac{\partial \sigma_{\varphi \varrho }}{\partial \varphi }+\frac{\partial \sigma_{z\varrho }}{\partial z}+\left(\frac{1}{\varrho }\right)\left(\sigma_{\varrho \varrho }-\sigma_{\varphi \varphi }\right)=0,\\ \frac{\partial \sigma_{\varrho \varphi }}{\partial \varrho } & +\left(\frac{1}{\varrho }\right)\frac{\partial \sigma_{\varphi \varphi }}{\partial \varphi }+\frac{\partial \sigma_{z\varphi }}{\partial z}+\left(\frac{2}{\varrho }\right)\sigma_{\varrho \varphi }=0,\\ \frac{\partial \sigma_{\varrho z}}{\partial \varrho } & +\left(\frac{1}{\varrho }\right)\frac{\partial \sigma_{\varphi z}}{\partial \varphi }+\frac{\partial \sigma_{zz}}{\partial z}+\left(\frac{1}{\varrho }\right)\sigma_{\varrho z}=0,\\ \frac{\partial D_{\varrho }}{\partial \varrho } & +\left(\frac{1}{\varrho }\right)D_{\varrho }+\frac{\partial D_{z}}{\partial z}=0. \end{array}$$

These equations are invariant with respect to rotations around the *z* axis (in spite of the lack of axisymmetry of the underlying wurtzite lattice [39]); hence solutions can be separated into a (ϱ,z) part and a φ part. If we apply these equations to a cylindrical symmetric wurtzite quantum dot, with adequate boundary conditions, the axisymmetry of the system (equations and geometry) leads to the absence of angular dependence, therefore $u_{\varphi }=0$. We can write the remaining equations in the following instructive form:(70)$$\begin{array}{c} LU=f, \end{array}$$
where $U\equiv (u_{r},u_{z},V)$, *L* is a second-order differential operator given by:(71)$$\begin{array}{c} L=\left(\begin{array}{ccc} \frac{\partial }{\partial \varrho }C_{11}\frac{\partial }{\partial \varrho }+\frac{\partial }{\partial z}C_{44}\frac{\partial }{\partial z}+\frac{1}{\varrho }\frac{\partial }{\partial \varrho }C_{12}+C_{11}\frac{\partial }{\partial \varrho }\frac{1}{\varrho } & \frac{\partial }{\partial \varrho }C_{13}\frac{\partial }{\partial z}+\frac{\partial }{\partial z}C_{44}\frac{\partial }{\partial \varrho } & \frac{\partial }{\partial \varrho }e_{31}\frac{\partial }{\partial z}+\frac{\partial }{\partial z}e_{15}\frac{\partial }{\partial \varrho }\;\\ \frac{\partial }{\partial \varrho }C_{44}\frac{\partial }{\partial z}+\frac{\partial }{\partial z}C_{13}\frac{\partial }{\partial \varrho }+\partial_{z}C_{13}\frac{1}{\varrho }+C_{44}\frac{1}{\varrho }\frac{\partial }{\partial z} & \frac{\partial }{\partial \varrho }C_{44}\frac{\partial }{\partial \varrho }+\frac{\partial }{\partial z}C_{33}\frac{\partial }{\partial z}+C_{44}\frac{1}{\varrho }\frac{\partial }{\partial \varrho } & \frac{\partial }{\partial \varrho }e_{33}\frac{\partial }{\partial z}+\frac{\partial }{\partial z}e_{15}\frac{\partial }{\partial \varrho }+e_{15}\frac{1}{\varrho }\frac{\partial }{\partial \varrho }\\ \frac{\partial }{\partial \varrho }e_{15}\frac{\partial }{\partial z}+e_{15}\frac{1}{\varrho }\frac{\partial }{\partial z}+\frac{\partial }{\partial z}e_{31}\frac{\partial }{\partial \varrho }+\frac{\partial }{\partial z}e_{15}\frac{1}{\varrho } & \frac{\partial }{\partial \varrho }e_{15}\frac{\partial }{\partial \varrho }+e_{15}\frac{1}{\varrho }\frac{\partial }{\partial \varrho }+\frac{\partial }{\partial z}e_{33}\frac{\partial }{\partial z} & -\frac{\partial }{\partial \varrho }{\hat{\varepsilon }}_{11}\frac{\partial }{\partial \varrho }-\frac{\partial }{\partial z}{\hat{\varepsilon }}_{33}\frac{\partial }{\partial z}-{\hat{\varepsilon }}_{11}\frac{1}{\varrho }\frac{\partial }{\partial \varrho } \end{array}\right), \end{array}$$
and *f* is the source term given by
(72)$$f=\left(\begin{array}{c} -\frac{\partial }{\partial \varrho }\left[\left(C_{11}+C_{12}\right)a+C_{33}c\right]\\ -\frac{\partial }{\partial z}\left[2C_{13}a+2C_{13}c\right]\\ -\frac{\partial }{\partial z}P_{sp} \end{array}\right).$$

Because the functions appearing in the source term are piecewise constant what we in reality have are surface sources for the strain and the electric potential (the discontinuities appear at the interfaces).

## 3. The Valence Force Fields Model

The atomistic valence force field model was first applied to study the lattice dynamics of diamond by Musgrave and Pople [20]. Later, Nusimovici and Birman [21] developed a model for wurtzite semiconductors with eight adjustable parameters. The most popular valence force fields scheme is probably due to Keating [22], where two parameters α and β are used. This model applies to covalent semiconductors.

From an atomistic point of view we can divide the requirements of an elastic strain energy $F_{s}$ into two groups:General conditions: rotational and displacement invariance.Conditions imposed by the symmetry of the crystal structure.

For the sake of simplicity we consider any general type of deformation and we assume that the elastic strain depends only on the positions of the nuclei. The latter is only valid in nonmetallic crystals, since in this case the Born-Oppenheimer approximation ensures that the electrons completely follow the nuclei. So the following models of valence force fields hold only with the condition that the forces on the electrons are always negligible [40]. The requirement that the energy is invariant under any arbitrary displacement of the lattice is satisfied if $F_{s}$ depends only on the difference between nuclear positions:(73)$$\begin{array}{c} F_{s}=F_{s}\left({\vec{x}}_{k}-{\vec{x}}_{l}\right)=F_{s}\left({\vec{u}}_{kl}\right), \end{array}$$
where ${\vec{u}}_{kl}={\vec{x}}_{k}-{\vec{x}}_{l}$ and ${\vec{x}}_{k}$ is the position vector of the *k*th nucleus after deformation. However, $F_{s}$ must be invariant under a transformation in which the atoms are displaced by a rigid rotation of the crystal, and ${\vec{u}}_{kl}$ are not invariant under such a transformation, since they transform as vectors. We can form an invariant from scalar products of ${\vec{u}}_{kl}$ and functions of such products:(74)$$\begin{array}{c} \lambda_{klmn}=\left({\vec{u}}_{kl}\cdot {\vec{u}}_{mn}-{\vec{U}}_{kl}\cdot {\vec{U}}_{mn}\right)/2a, \end{array}$$
where *a* is the lattice constant, ${\vec{U}}_{kl}={\vec{X}}_{k}-{\vec{X}}_{l}$ and ${\vec{X}}_{k}$ is the position vector the *k*th nucleus in the undeformed crystal. The final term is included so that the invariant is equal to zero when the deformation is removed. The energy $F_{s}$ is a function of a large number of $\lambda_{klmn}$, and since these are small, we can use them as a basis for a series expansion of $F_{s}$. We disregard the constant term, while the linear terms vanish if the energy is an extremum at equilibrium. So for small strain we can write using the summation convention:(75)$$\begin{array}{c} F_{s}=\frac{1}{2}B_{klmn}^{pqrs}\lambda_{klmn}\lambda_{pqrs}+O\left(\lambda^{3}\right). \end{array}$$

The coefficients $B_{klmn}^{pqrs}$ must be positive definite in order to ensure that $F_{s}$ is a definite minimum. We still have more terms $\lambda_{klmn}$ than necessary because most of the coefficients $B_{klmn}^{pqrs}$ are not independent. It was shown that there are only 3N−6 independent invariants [22], which have to be determined from the crystal structure.

We consider a slightly deformed primitive structure, which we can figure as a large number of parallelepipeds with atoms at each of the eight corners. Without deformations, all parallelepipeds are identical unit cells. The arrangement of the eight atoms on the corners of a cell is given by 18 scalar products, and a convenient set of these is obtained by taking the squares of the lengths of the 12 edges of the cell and the 6 off-diagonal products, i.e., the angles between vectors, represented by arcs in Figure 2.

The four atoms of the adjacent cell which are not already fixed by the above scalar products are determined by the eight remaining edge lengths of this cell and by four more angles. The rest of the crystal is included by adding cells and using only the necessary scalar products for the atom positions. We have three different types of lattice points according the number of necessary scalar products. First, the points lying along three lines passing through point 0 (see again Figure 2) in the initial cell and parallel to the three basis vectors of the undistorted lattice, which are associated with three diagonal products (edge lengths) and three off-diagonal products (angles), as shown in Figure 3a.

Second, we have the points in the reference planes, but off the reference lines, associated with three diagonal products but only one off-diagonal product, as shown in Figure 3b. Finally, we have the points which do not lie in the above mentioned planes, associated with three diagonal scalar products but no off-diagonal products, as shown in Figure 3c. This nonuniformity in the distribution of the scalar products is clearly undesirable, and we can remove it by invoking the invariance of a crystal under the operations of the relevant translation subgroup, and by the assumption that interactions over distances of the order of the crystal dimensions are negligible. This assumption is necessary, especially if we are not considering a bulk material, such as in a heterostructure quantum dot.

We write ${\vec{x}}_{1}\left(l\right),{\vec{x}}_{2}\left(l\right),{\vec{x}}_{3}\left(l\right)$ as the position vectors of three neigbors of the atom cell (l) relative to this latter atom and which become the lattice basis vectors when the distortion is removed; so that we can rewrite (Equation 75) in the following way:(76)$$\begin{array}{c} F_{s}=\frac{1}{2}\sum_{l,l^{\prime }}\sum_{m,n,m^{\prime },n^{\prime }=1}^{3}B_{mnm^{\prime }n^{\prime }}\left(l-l^{\prime }\right)\lambda_{mn}\left(l\right)\lambda_{m^{\prime }n^{\prime }}\left(l^{\prime }\right)+\cdots , \end{array}$$
where
(77)$$\begin{array}{c} \lambda_{mn}\left(l\right)=\left({\vec{x}}_{m}\left(l\right)\cdot {\vec{x}}_{n}\left(l\right)-{\vec{X}}_{m}\cdot {\vec{X}}_{n}\right)/2a, \end{array}$$
and is symmetric in (m,n), the sums over $l,l^{\prime }$ run over all the unit cells, and $B_{mnm^{\prime }n^{\prime }}\left(l-l^{\prime }\right)$ is invariant under all the operations of the space group, it is positive, defined and falls off rapidly as $\left(l-l^{\prime }\right)$ increases, in order to have a convergent expression.

We can extend this formulation to the case of nonprimitive structures: for a diatomic structure one suitable set of scalar products consist of a set, just as the previous one, but using the atoms on one sublattice together with three extra scalars per unit cell of the diatomic structure is necessary to locate the *B* atom relative to the *A* atom. However, a more convenient set is given by using ${\vec{x}}_{1}\left(l\right),{\vec{x}}_{2}\left(l\right),{\vec{x}}_{3}\left(l\right)$ as the position vectors of the *B* atoms in the neighboring unit cells relative to the *A* atom of cell (l) and ${\vec{x}}_{4}\left(l\right)$ as the position vectors of the atom *B* in the cell (l) relative to the *A* atom there. Thus, we can write $F_{s}$ as:(78)$$\begin{array}{c} F_{s}=\frac{1}{2}\sum_{l,l^{\prime }}\sum_{m,n,m^{\prime },n^{\prime }=1}^{4}B_{mnm^{\prime }n^{\prime }}\left(l-l^{\prime }\right)\lambda_{mn}\left(l\right)\lambda_{m^{\prime }n^{\prime }}\left(l^{\prime }\right)+\cdots . \end{array}$$

In the same way it is possible to extend the formulation to structures with a greater number of atoms per unit cells. Within the harmonic approximation Equation (Equation 78) becomes:(79)$$\begin{array}{c} F_{s}=\frac{1}{2}K_{ab}^{mn}\left(l-l^{\prime }\right)u_{a}^{m}u_{b}^{n}, \end{array}$$
where $u_{a}^{m}$ is the *a*th component of the displacement of the *m*th nucleus and the $K_{ab}^{mn}$ are linear combinations of the $B_{mnm^{\prime }n^{\prime }}$, and this form is suitable for calculation purposes, and it has been applied to the calculation of the elastic constants of the diamond structure [22]. This model includes only two types of interaction: A nearest-neighbor term and a noncentral second-neighbor term. The basic unit cell of the diamond structure is a rhombohedron with two atoms (atoms 1 and 0 in Figure 4) on its major axis, which is directed along the [111] direction. The three neighboring unit cells of interest contain atoms 2 and 5, 3 and 6, 4 and 7, respectively.

The following expression for the strain energy with two constants is derived from Equation (Equation 79) by including only diagonal products of the λs:(80)$$\begin{array}{cc} F_{s} & =\frac{1}{2}\sum_{l}\sum_{m,n=1}^{4}B_{mnmn}\left(O\right)\lambda_{mn}^{2}\left(l\right)\\ \; & =\frac{1}{2}\sum_{l}\left[\frac{\alpha }{4a^{2}}\sum_{i=1}^{4}{\left(x_{0i}^{2}\left(l\right)-3a^{2}\right)}^{2}+\frac{\beta }{2a^{2}}\sum_{i,j>i1}^{4}{\left(x_{0i}\left(l\right)\cdot x_{0j}\left(l\right)+a^{2}\right)}^{2}\right], \end{array}$$
where the atomic labeling is as in Figure 4 and the required symmetry has been imposed by $B_{mmmm}\left(O\right)=\alpha$ (for all *m*), $B_{mnmn}\left(O\right)=\beta$ (all m,n,m≠n), and by including the term in $\lambda_{34}^{2}$ only if the symmetry $B_{3434}=B_{1212}$, etc., is satisfied.

Successively physical properties of semiconductor alloy $A_{1-x}B_{x}C$ have been studied using a valence force fields model [41]. Lattice-mismatched zincblende semiconductor alloy ground state configurations have been determined [42], and also the groundstate search of a group of lattice-mismatched III-V semiconductor alloys, such as GaInN, GaInP, GaInAs, GaInSb, InAsSb, and InPAs has been performed. A valence force fields model for lattice-mismatched isovalent semiconductor zincblende alloys has been derived in Ref. [41], where the strain energy was given by:(81)$$\begin{array}{cc} \;\;\;\;\;\;\;\;F_{VFF} & =\sum_{l}\sum_{m=1}^{4}\frac{3\alpha_{lm}}{8d_{lm}^{2}}{\left[r_{lm}^{2}-d_{lm}^{2}\right]}^{2}\\ \; & +\sum_{l}\sum_{s=1}^{2}\sum_{k=1}^{6}\frac{3\beta_{lsk}}{8d_{lsk1}d_{lsk2}}{\left[r_{lsk1}r_{lsk2}\cos\left(\Theta_{lsk}\right)-d_{lsk1}d_{lsk2}\cos\left(\Theta_{0}\right)\right]}^{2}, \end{array}$$
where *l* runs through all the lattice sites in the unit cell, s=1,2 denotes the two sublattice sites in in the zincblende cell, *m* runs through the four different bonds, and *k* runs through the six angles with the vertex at site ls. The two bonds that form the angle *k* at the site ls are represented by lsk1 and lsk2, while $d_{lm}$ (similarly for $d_{lsk1}$ and $d_{lsk2}$) is the ideal bond length for bond lm, and $r_{lm}$ (similarly for $r_{lsk1}$ and $r_{lsk2}$) is the corresponding calculated bond length. The angle formed between lsk1 and lsk2 is given by $\Theta_{lsk}$, while $\Theta_{0}=109.5^{{}^{\circ}}$ is the ideal tetrahedral bond angle.

Martin [24] utilized bond-stretching and (α) and bending (β) parameters similar to Keating but added point-ion Coulombic forces to the free energy. We used the model given by Equation (Equation 81) to develop the Keating model for wurtzite quantum dot heterostructures. The free energy of the elastic part is given by a sum over all atoms *i*:(82)$$\begin{array}{c} \;\;\;\;\;\;\;\;F_{VFF}=\sum_{i}\left(\sum_{j}\left[\frac{3\alpha_{ij}}{8d_{ij}^{2}}{\left(r_{ij}\cdot r_{ij}-d_{ij}^{2}\right)}^{2}\right.\right.+\left.\left.\sum_{k\neq j}\frac{3\beta_{ijk}}{8d_{ij}d_{ik}}{\left(r_{ij}\cdot r_{ik}-d_{ij}d_{ik}\cos\left(\Theta_{ijk}\right)\right)}^{2}\right]\right), \end{array}$$
and the sums over *j* and *k* run over the nearest neighbor atoms, *d* and *r* are the bulk and distorted distances between neighbor atoms, $\Theta_{ijk}$ is the ideal unrelaxed tetrahedral bond length, and α,β are empirical material-dependent elastic parameters as mentioned above. At present, the piezoelectric effect has not been included in the energy expression for the valence force fields model, but we will demonstrate how it can be incorporated using a semi-coupled model.

### 3.1. Calculation of a Strain Tensor from Atomistic Data

For each atom, there exists a displacement vector, but in the discrete atomic case, the classical continuous strain tensor relies on derivatives of the displacement vector, which lack well-definedness. To overcome these challenges, two approaches are available for strain tensor calculations. One approach involves determining the displacement vectors for each atom and employing Shepard’s interpolation [43] to compute a continuous displacement vector that enables differentiation for strain tensor calculation. To account for lattice-mismatch, it is manually subtracted from the diagonal components when inside the dot. The other approach follows the methodology presented in Ref. [44], where a discrete deformation gradient is defined through an optimization procedure. Lattice mismatch is automatically incorporated when utilizing the unstrained bond lengths of the dot material inside the dot.

Generally, the two methods yield similar results for the strain tensor components. However, the interpolation method often exhibits more oscillations and irregularities compared to the atomic calculation. The atomic method is computationally simpler and faster, and we primarily employ this approach. The oscillations and irregularities observed in the interpolation method are highly dependent on the chosen trivariate interpolation method.

### 3.2. Inclusion of Piezoelectric Effect in Valence Force Field Calculation

The VFF model usually does not incorporate the piezoelectric effect, as it solely accounts for nearest neighbor interactions. However, it is possible to include the piezoelectric effect in a semi-coupled manner by calculating the electric potential using Gauss’s law:(83)∇·D=0,
where
(84)$$D_{i}=-\varepsilon_{in}\frac{\partial V}{\partial x_{i}}+e_{ilm}\epsilon_{lm},$$

Here, $\epsilon_{lm}$ represents interpolated strain fields based on VFF calculations. This semi-coupled approach signifies that the electric field is coupled to the strain fields, while the strain fields remain independent of the electric field. Previous research has demonstrated the effectiveness of the semi-coupled approach for most zincblende compounds, including the very relevant InAs/GaAs system [45].

## 4. How to Connect Continuum Model and VFF

For bulk materials, where $d_{ij}$, $\alpha_{ij}$, and $\beta_{ij}$ are independent of atomic index, the parameters α and β can be expressed in terms of $C_{11}$ and $C_{12}$ as follows [46]:(85)$$\begin{array}{ccc} \alpha & = & \frac{1}{\sqrt{3}}\left(C_{11}+3C_{12}\right)d\;\;\;\mathrm{and} \end{array}$$(86)$$\begin{array}{ccc} \beta & = & \frac{1}{\sqrt{3}}\left(C_{11}-C_{12}\right)d. \end{array}$$

In the VFF model, the parameter $C_{44}$ can be calculated from α and β as shown in [47]:(87)$$C_{44}^{\mathrm{VFF}}=\frac{\sqrt{3}\alpha \beta }{(\alpha +\beta )d}.$$

At the interface between a matrix and a dot material, an average β-value is required for the alloy combination. Traditional elasticity does not exhibit inherent size dependence in the elastic solutions of embedded inhomogeneities. In systems with dimensions larger than 50 nm, the surface-to-volume ratio is usually insignificant, and the deformation behavior is primarily governed by classical bulk strain energy. Presently, there is no available framework that combines interface and surface elasticity with bulk elasticity to analyze embedded inclusions. The approach utilized in quantum dot literature relies solely on classical bulk elasticity. For more information on corrections related to hydrostatic strain arising from interfacial and surface elasticity, please refer to Refs. [48,49].

## 5. Boundary Conditions

Quantum dots are typically surrounded by a matrix material that is much larger in size, allowing us to consider it as having infinite extent. However, when solving the numerical equations, it is necessary to define a finite computational domain. Consequently, boundary conditions must be imposed on this computational domain to ensure that the electromechanical field near the dot is not influenced by these artificial boundaries.

### 5.1. Boundary Conditions in Continuum Model

In the case of the continuum model, we can mention four main types of boundary conditions:1.Fixed boundary conditions, which enforce a zero displacement vector at the boundary:
(88)$$\vec{u}{|}_{\partial \Omega }=0,$$
where ∂Ω is the boundary of the computational domain Ω.2.Free boundary conditions, which imply the absence of forces at the boundary and therefore the surface traction vector is required to be zero. The surface traction vector is given by
(89)$${\left[T_{n}\right]}_{i}=\sum_{j=1}^{3}\sigma_{ij}n_{j},$$
where $\vec{n}$ is the outward pointing unit normal vector to the surface. The free boundary condition is then give by
(90)$${\vec{T}}_{n}{|}_{\partial \Omega }=\vec{0}.$$3.Periodic boundary conditions that simulate a periodic arrangement of quantum dots with a wetting layer. There is limited existing literature on the determination of boundary conditions for strain aimed at modeling a periodic arrangement of a particular structure. This task is non-trivial since the displacement vector is not necessarily periodic, as the structure should be allowed to expand in all directions even within an array of quantum dots. A possible option is to establish a set of periodic boundary conditions on a single cell of the periodic structure (referred to as the domain) based on the following two principles.The first principle is based on Newton’s second law, which states that in a static scenario, the sum of all forces should be zero. For a periodic array, this implies that the traction force from one side of the domain must be equal in magnitude and opposite in direction to the traction force from the other side of the domain, expressed as:
(91)$${\vec{T}}_{n}\left|\partial \Omega_{1}=-{\vec{T}}_{n}\right|\partial \Omega_{2},$$
where $\partial \Omega_{1}$ and $\partial \Omega_{2}$ represent the periodic boundaries.The second principle ensures that no cracks are allowed to appear on the surface of the domain. This requirement is satisfied when:
(92)$$\frac{\partial \vec{n}\cdot \vec{u}}{\partial \xi_{i}}\left|\partial \Omega_{1}=-\frac{\partial \vec{n}\cdot \vec{u}}{\partial \xi_{i}}\right|\partial \Omega_{2},$$
where $\left(\xi_{1},\xi_{2}\right)$ parameterize the surface, $\xi_{3}$ parameterizes the direction normal to the surface, and i=1,2,3. The minus sign accounts for both normal vectors pointing outward from the domain. The condition for i=1,2 ensures that the slopes of the two surfaces are equal, and the final condition ensures that the normal derivative of the displacement vector, in the direction normal to the boundaries, is equal at periodic boundaries. These boundary conditions have been verified in the two-dimensional case in the study by Lassen et al. [50].4.Fixed free boundary conditions for quantum dots with wetting layers. It is generally acknowledged that in the case of a quantum well, the nearly infinite extent of the matrix material imposes constraints on the expansion of the system perpendicular to the well. However, in the direction of the quantum well, the material is allowed to expand. This expectation also applies to a truncated pyramid with a wetting layer. Therefore, a more suitable set of boundary conditions would involve setting the displacement to zero in the perpendicular direction to the wetting layer at the boundaries, while maintaining the remaining boundary conditions as free.

### 5.2. Boundary Conditions in Valence Force Field

The Keating energy (Equation 81) accounts for all atoms as well as their nearest neighbors. Interior atoms in InAs are connected to four neighboring atoms, while the number of bonds for atoms at the computational domain’s boundary depends on the chosen boundary conditions. One option is to consider dangling bonds, where atoms on the boundary can have zero to four bonds depending on their position (atoms with zero bonds can be entirely removed from the system). Another approach involves introducing artificial boundary atoms to ensure that atoms on the boundary always have four bonds, although some of these bonds are formed with artificial atoms. The question then arises regarding the selection of parameters for these artificial atoms. One possibility is to divide all α and β parameters by two, as a sort of one-step interpolation towards a vacuum. The third option is to employ periodic boundary conditions by defining a computational box and requiring that all atoms have four bonds. Bonds extending beyond the computational box will wrap around and re-enter from the opposite side of the box. The energy minimum becomes dependent on the size of the box, and in most cases, it is desirable to also minimize the energy with respect to the box size and, possibly, its shape. This can be achieved by introducing a spatially constant metric tensor, expressing the atom coordinates in terms of the metric, and optimizing the metric tensor components as well [51].

## 6. GaN/AlN Wurtzite Quantum Dots

### 6.1. A Fully-Coupled Continuum Model for GaN/AlN Wurtzite Cylindrical Quantum Dot

The governing equations for the electromechanical fields of wurtzite structures are axisymmetric, hence all electric- and mechanical-field solutions are axisymmetric as well and the original three-dimensional problem can be solved as a two-dimensional mathematical-model problem [38,52,53]. In order to check the validity of the cylindrical model of Section 2.8, the complete three-dimensional fully coupled model given by Equation (Equation 48) for an ideal cylindrical wurtzite GaN/AlN quantum dot with radius r=6 nm and height h=3 nm has been solved.

The strain tensor $\varepsilon_{zz}$ calculated by the two-dimensional model (top) and the three-dimensional model (bottom) is plotted in Figure 5. The plot for the three-dimensional model shows a slide of the quantum dot in the xy plane, correspondent to the double of the surfaces in the r,z plane given by the rotational-invariant model in two dimensions. Not only is an almost perfect qualitative agreement of the strain field in the two plots observable, but there is also an excellent agreement of the maximum and minimum values of $\varepsilon_{zz}$ inside of the dot and in the matrix.

With the three-dimensional model we can also verify the cylindrical symmetry of the electric field *E* for a wurtzite cylindrical quantum dot in the xy plane, showing in Figure 6 the absolute value of *E* on this plane.

### 6.2. A GaN/AlN Wurtzite Hexagonal Pyramid Quantum Dot

In this section three different hexagonal quantum dots with wetting layer (refer to Figure 7 for parameter meanings and geometry) are studied. The dimensions of the dots are given in Table 1. The first two dots (Dot 1 and 2) are narrow dots with a bottom diameter of 4.936 nm and the last (Dot 3) is wide having a bottom diameter of 8.638 nm.

In the first three rows of Figure 8 the strain components $\varepsilon_{xx}$ and $\varepsilon_{zz}$ and the electric potential *V* for the three dots along the center of the structures are shown. There is a clear qualitative agreement for the $\varepsilon_{xx}$ and $\varepsilon_{zz}$ strain components between the three models: VFF, fully coupled, and semi-coupled continuum. However, quantitative differences exist, locally up to approximately 25%, between VFF and continuum models. The difference between a fully and semi-coupled model is that in the semi-coupled model, instead of solving the complete system of four differential coupled equations, we disregard the piezoelectric field and spontaneous polarization, and we solve three equations only for the strain fields, and successively we calculate the piezoelectric potential from the given strain fields. Further, these results demonstrate good quantitative agreement between semi-coupled and fully coupled continuum data (locally up to maximum 5%). Similarly, for the electric potential a notable difference between the VFF and the continuum results is observed. The differences in the electric field are a direct consequence of the differences in the strain fields. Since VFF parameters are computed using non-piezoelectric corrected stiffness coefficients, better agreement between semi-coupled continuum and VFF vs. fully-coupled continuum and VFF is expected. This is indeed confirmed by the results in Figure 8. Deviations between VFF and continuum results for the electric potential are mainly due to discrepancies in the off-diagonal strain components $\varepsilon_{ij}\left(i\neq j\right)$ (see Figure 9).

In Table 2 the electronic groundstate energies for the three dots found using the effective mass approximation for the conduction band are shown [54,55]. The effect of strain and electric field has been included via the effective potential [25]: $V_{eff}=a\left(\varepsilon_{xx}+\varepsilon_{yy}\right)+b\left(\varepsilon_{zz}\right)+eV+E_{edge}$. Here *a* and *b* are deformation potentials, *e* is the electronic charge, and $E_{edge}$ is the bulk band edge. The effective potential for the three dots is shown in row four of Figure 8. Firstly, a difference between the VFF and the continuum results of up to 100 meV is observed. Secondly, a smaller difference between the fully coupled and the semi-coupled models for dots 1−3 of up to 15 meV is found. The differences in the effective potential are responsible for the variations in the groundstate energies observed in Table 2.

## 7. InGaN Wurtize Quantum Dots

It is widely recognized that InGaN alloys have now become the fundamental materials for the active regions of visible light-emitting diodes (LEDs), which are playing an increasingly significant part in the global advancement of solid-state illumination [56,57,58,59].Hence, it is highly significant to investigate the influence of electromechanical coupling on the optical characteristics of light-emitting diodes (LEDs) featuring InGaN/GaN quantum-dot (QD) active regions using computational modeling. In this approach, the configuration, including the morphology and the mean In composition of the QDs, has been directly deduced from experimental data on the distribution of strain perpendicular to the growth plane, acquired through geometric-phase analysis of a high-resolution transmission electron microscopy (HRTEM) image of an LED structure fabricated via metalorganic vapor-phase epitaxy (MOVPE).

A series of LED architectures were fabricated on (0001) sapphire substrates using metalorganic vapor-phase epitaxy (MOVPE) in an AIX2000HT system. These structures were comprised of an unintentionally-doped (UID) GaN buffer layer, a thick n-GaN contact layer with a dopant concentration of $5\times 10^{18}{\mathrm{cm}}^{-3}$, a 12-period short-period superlattice (SPSL) of ${\mathrm{In}}_{0.1}{\mathrm{Ga}}_{0.9}N$ (1 nm)/GaN (1 nm) synthesized via conversion [60], a low-temperature UID GaN layer (20 nm), a UID active region (AR) containing two stacked layers of InGaN quantum dots (QDs) separated by a GaN spacer (7 nm), an UID GaN barrier layer (4 nm), a $p-{\mathrm{Al}}_{0.1}{\mathrm{Ga}}_{0.9}N$ electron blocking-layer (EBL) (12 nm) doped with [Mg] = $5\times 10^{19}{\mathrm{cm}}^{-3}$, and a p-GaN contact layer (180 nm) doped with [Mg] = $5\times 10^{19}\;{\mathrm{cm}}^{-3}$.

The formation of QDs within the structures was achieved in-situ during growth by employing a method that involved interrupting the growth process in a mixed nitrogen/hydrogen atmosphere at moderate pressure after depositing a thin layer of InGaN. The introduction of hydrogen during the growth resulted in local etching of the InGaN quantum well (QW) and the formation of distinct islands [60]. Dark field TEM imaging was employed to assess the structural quality and provide an overall visualization of the LED structures at a moderate level of magnification. The experiments were conducted using a Jeol 2010 microscope operating at 200 kV. Geometric Phase Analysis (GPA) [61] was applied to high-resolution transmission electron microscopy (HRTEM) images for mapping the strain relative to the reference GaN lattice within the active regions (ARs). This technique enabled the extraction of alloy composition and thickness variations within the InGaN quantum wells (QWs) with a subnanometer spatial resolution.

The experiments were carried out on a SACTEM-Toulouse microscope (Tecnai-FEI) operating at 200 kV and equipped with an image aberration corrector. HRTEM images were acquired along the [5-4-10] zone axis, capturing the (0002) planes exclusively. Analysis of the HRTEM images was performed using a GPA Phase 3.5 (HREM Research Inc., Tokyo, Japan) and plugins within the Digital Micrograph image processing package (Gatan Inc., Pleasanton, CA, USA). The estimated spatial resolution of the HRTEM images was approximately ∼0.5–1.0 nm. Figure 10 depicts a strain map relative to GaN, denoted as $\varepsilon_{zz}^{GaN}$, obtained with a spatial resolution of 0.7 nm in the examined LED structure. The map reveals a portion of the InGaN/GaN short-period superlattice (SL), including two InGaN quantum dot (QD) layers separated by a GaN spacer, the final GaN barrier, the AlGaN electron-blocking layer (EBL), and a segment of the p-GaN contact layer. The image contrast represents variations in the out-of-plane $\varepsilon_{zz}^{GaN}$ strain component. In this image, the InGaN layers are depicted in shades of red and yellow, while the EBL appears in green. The QDs exhibit heights of approximately 3 nm and lateral dimensions ranging from about 50 to 100 nm. Notably, all the QDs exhibited non-uniform Indium content distribution, ranging from approximately 16% to 23%, with typical lateral variations occurring on the scale of a few nanometers.

The emission spectra of single QDs can be significantly altered by inter-dot strain fields, not only in the case of vertical coupling where multiple dots are vertically aligned, but also in the case of lateral coupling where closely spaced QDs are grown on the same plane [62]. In this study, the distribution of the strain field in the active region was examined using a simulation with a continuum model. Figure 11 illustrates the magnitude of the hydrostatic strain field on the yz plane [63,64,65].

It is worth noting that the strain exhibits significant intensity not only within the quantum dots, as expected, but also between the two layers of dots, while rapidly diminishing beyond the dots.

## 8. InAs/GaAs Zincblende Quantum Dots

### 8.1. A InAs/GaAs Zincblende Cylindical Quantum Dot

In this case the three-dimensional Equation (Equation 47) for an ideal cylindrical quantum dot with radius r=6 nm and height h=3 nm, have also been solved. Looking at the plot of the strain fields $\varepsilon_{zz}$ and $\left(\varepsilon_{xx}+\varepsilon_{yy}\right)$ given by the anisotropic three-dimensional model as shown in Figure 12, it is still apparent that they show a sort of cylindrical symmetry, despite the cubic structure of a zincblende crystal [66].

It is important to mention in this context that solving a three-dimensional fully coupled or semi-coupled model gives almost the same results, i.e., a solution to Equation (Equation 47) including or excluding the piezoelectric effect affects the strain fields very slightly. This is mainly due to the fact that in a zincblende quantum dot the polarization is given by the off-diagonal components of the strain fields, as can be seen from Equation (Equation 40). This leads to a weak electric field almost all outside of the dot, as we can see from Figure 13, where it is shown as the absolute value of the electric field for this cylindrical quantum dot, given by the fully-coupled three-dimensional model.

It is interesting to compare this last result with the one given in the previous section in Figure for a GaN/AlN wurtzite quantum dot with the same shape and dimensions. Not only do the electric fields present a completely different symmetry, because of the different components which generate the piezoelectrical fields [see Equations (Equation 40) and (Equation 43)], but the maximum value in the zincblende case is almost 30 time smaller as well. A similar result has already been observed in [25].

### 8.2. An Inhomogeneous InAs/GaAs Quantum Dot

It is widely recognized that InAs/GaAs quantum dot structures, grown using the Stranski-Krastanow growth mode [67], exhibit a non-uniform indium concentration within the quantum dots [68]. Specifically, it has been observed that the indium concentration is highest at the apex of pyramidal quantum dots and decreases towards the base. The profile of indium concentration is crucial for accurately predicting electronic properties for two main reasons. Firstly, within the $\vec{k}\cdot \vec{p}$ approximation [69,70], the concentration directly impacts the effective masses and confinement potentials, resulting in energy shifts and changes in dipole transition strengths. Secondly, the concentration profile influences the strain distribution and, consequently, the piezoelectric potential, leading to additional energy shifts and alterations in transition strengths.

Various methods exist for determining concentration profiles in quantum dot structures, such as cross-sectional transmission electron microscopy (X-TEM) [71], cross-sectional scanning tunneling microscopy (X-STM) [72], scanning transmission electron microscopy (STEM) [73], X-ray photoelectron microscopy [74], anomalous X-ray scattering (AXS) [75], Scanning-Probe- Microscopy nanotomography [76], and composition evaluation by lattice fringe analysis (CELFA) technique [77]. Although the ultimate goal is to obtain information about the full three-dimensional concentration profile, current methods either provide average information within a given cross-section or give surface-level concentration profiles. Here we focus on extrapolating an average concentration profile obtained using the CELFA technique and investigate the implications of a specific extrapolation choice for a truncated pyramid quantum dot, as shown in Figure 14. We consider the following three different indium concentration profiles, which have been chosen with an identical total indium content.

Profile 1: we assume the indium concentration to be constant in the [010] direction.
(93)$$\begin{array}{c} C\left(x,y,z\right)=\left\{\begin{array}{cc} \hat{C}\left(x,z\right)/L\left(z\right) & \mathrm{inside}\;\mathrm{the}\;\mathrm{pyramid}\\ 0 & \mathrm{outside} \end{array}\right. \end{array}$$
where $L\left(z\right)=\left(L_{d}-2z\right)$ for 0<z<Ld/2 and zero otherwise, $\hat{C}$ is the average indium concentration, and $L_{d}$ is the side length of the pyramid at z=0. We divide by L(z) in order to take into account the shape of the quantum dot, see reference [77] for further details.

Profile 2: we assume a symmetric indium concentration profile in the [100] and [010] directions. Given an average concentration profile the three dimensional concentration profile takes the form
(94)$$\begin{array}{c} \hat{C}\left(x,z\right)=\frac{\hat{C}\left(x,z\right)\hat{C}\left(y,z\right)}{\int \hat{C}\left(x,z\right)dx} \end{array}$$
where again $\hat{C}$ is the average indium concentration. We divide with $\int \hat{C}\left(x,z\right)dx$ to have $\int \hat{C}\left(x,y,z\right)dx=\hat{C}\left(y,z\right)$.

Profile 3: We assume a constant indium concentration profile throughout the quantum dot.
(95)$$\begin{array}{c} C\left(x,y,z\right)=\left\{\begin{array}{cc} C_{tot}/V & \mathrm{inside}\;\mathrm{the}\;\mathrm{pyramid}\\ 0 & \mathrm{outside} \end{array}\right. \end{array}$$
where $C_{tot}=\hat{C}\left(x,z\right)dxdz$ and *V* is the volume of the pyramid.

In Figure 15, we present the hydrostatic strain component $H=\varepsilon_{xx}+\varepsilon_{yy}+\varepsilon_{zz}$ for the three indium concentration profiles [63]. The influence of the chosen indium concentration profile was clearly manifested in the hydrostatic strain. The most noticeable disparities were observed in regions where a constant indium concentration was assumed. For instance, when comparing profile 1 with profile 2, the primary distinctions emerged in the [010] direction. Furthermore, we observed that in profile 1, the strain reaches its peak at the apex of the pyramid, whereas in profile 2, elevated strains were also observed deeper within the structure due to a higher indium concentration in those regions.

### 8.3. A Truncated-Pyramid InAs/GaAs Quantum Dot

For InAs/GaAs quantum dots the truncated-pyramid with a wetting layer closely resembles experimental shapes of self-assembled Stranski-Krastanov InAs QDs [78,79]. In this context, the integration of continuum and atomistic methods for a multiscale depiction of a structure may be pertinent. Another intriguing approach involves incorporating components of the actual experimental structure into a multiscale model. Although it has been argued that an unspecified geometric irregularity of the quantum dots is necessary to reproduce the correct symmetry and energy level splitting within the $\vec{k}\cdot \vec{p}$ model [80], recent studies have indicated that a realistic replication of the strain distribution [81] and quantum confinement [62,82] in heterostructures can achieve comparable symmetries to an atomistic approach.

Nevertheless, when dealing with nanoscale systems, continuum models may fail in capturing the correct symmetries due to their lack of an atomistic structure description [83]. Additionally, the continuum model fails to accurately depict interface and surface features on an atomistic scale, leading to a lack of information regarding internal strain [4]. The consideration of internal strain is crucial for obtaining atomistic solutions for band structures, as observed in tight binding methods. Therefore, the continuum elastic model can serve as an initial estimate for a Keating Valence Force Field relaxation [22], as detailed in Ref. [84]. The parameters (α and β) employed for the VFF calculations are selected to ensure consistency with the results obtained from the continuum model, where elastic constant relations for zincblende crystals are utilized.

The piezoelectric field, which is incorporated directly into the fully-coupled continuum model, is computed using a semi-coupled model in the atomistic scenario, as Keating’s VFF model does not consider long-range Coulomb effects. Previous research has demonstrated that the disparities between the two approaches are insignificant for zincblende InGaAs structures (for further insights, please refer to Ref. [63] for a thorough examination). Therefore, we consider a truncated pyramid quantum dot with a base side length of 9 nm and a height of 2 nm. The need to incorporate the VFF method in atomistic band structure computations, despite the similarity in strain tensors between the continuum and atomistic methods, appears to suggest a significant impact of internal strain on the underlying crystal structure. To emphasize this, the internal strain on As atoms is represented as follows: The strain tensors are computed using a continuum model and then the outcomes are interpolated to determine the atomic positions within the strained crystal structure.

Given an As atom in position $r_{0}$ the bond lengths of the strained crystal structure calculated by a continuum model are defined as: (96)$$\begin{array}{c} d_{i}=r_{i}-r_{0}\;\mathrm{with}\;i=1,\ldots ,4, \end{array}$$
where $r_{i}$ is the position of one of the four nearest neighbors of the As atom. If $d_{j}$ is defined as the corresponding bond lengths obtained from the preconditioned VFF algorithm calculations on the continuum model results, the standard deviation of the bond lengths can be expressed after the application of the VFF algorithm for each individual As atom as: (97)$$\begin{array}{c} \sigma =\sqrt{\sum_{\left(i\neq j\right)}^{\;}\frac{{\left(d_{i}-d_{j}\right)}^{2}}{6}}\;\mathrm{with}\;i,j=1,\ldots ,4, \end{array}$$
where six are all the possible couples of four bonds. In Figure 16, σ is plotted on each individual As atom on the xz plane, both when considering only a continuum model and when incorporating VFF as well. It can be observed that VFF has a tendency to equalize the bonds, resulting in a decrease in differences. However, a gap still exists near the interfaces. This can be understood due to the expected deviation of the tetrahedron, as an As atom is connected to two Ga atoms on one side and two In atoms on the other.

## 9. InP Lateral Quantum-Dot Molecules

A very interesting aspect is the study of the impact of strain on quantum dot molecules is that the interaction between the dots gives rise to new energy levels similar to those found in molecules, including bonding and antibonding states. The most common arrangement is the vertical configuration, where multiple dots are stacked on top of each other with a thin layer of wider band-gap material separating them. Another configuration is the lateral coupling, where the dots are grown closely spaced on the same plane, allowing for the formation of molecular-like confined states. These systems hold great promise for the development of novel devices such as light emitting diodes (LEDs) and lasers, taking advantage of the emergence of molecular states in the radiative recombination within the infrared (IR) and terahertz (THz) ranges.

We consider as an example InP self-assembled quantum dots (SQDs) fabricated using epitaxial growth on an In${}_{0.48}$Ga${}_{0.52}$P buffer layer that was lattice-matched to a GaAs substrate doped with silicon. The synthesized SQDs were then characterized using atomic force microscopy (AFM) and continuous-wave photoluminescence techniques [82]. Two different quantum dot molecules (QDMs) were isolated from a homogeneous sample, and their three-dimensional structures were determined using the Gwyddion${}^{\mathrm{TM}}$ software [85]. Through sampling and extrapolation, the structures required for calculations were obtained. The first QDM comprises two dots that were nearly identical in size, with approximate dimensions of $R_{eq}=12.5$ nm and H=6.5 nm. In contrast, the second QDM consisted of two dots with significantly different sizes: A smaller dot with $R_{eq}=10.0$ nm and H=5.0 nm, and a larger dot with $R_{eq}=18.0$ nm and H=7.5 nm.

In accordance with the common analogy that defines a quantum dot (QD) as an artificial atom and, consequently, a quantum dot molecule (QDM) as an artificial molecule, the first QDM was designated as a homonuclear molecule (HO), while the second QDM was classified as a heteronuclear molecule (HE). These definitions will be employed throughout our paper. Figure 17 displays a plot of the extrapolated 3D structure of the homonuclear QD molecule.

To further analyze the QDMs, we constructed a finite element model (FEM) based on the extrapolated structures. This FEM serves as the foundation for the electro-mechanical models implemented and solved using the TiberCAD [86,87] simulator. In Figure 18, a top view of a three-dimensional plot illustrating the magnitude of the hydrostatic strain field for both QDMs is presented [63].

The comparison between the two figures revealed several notable similarities as well as interesting differences. In both cases, the strain field exhibited significant intensity along the outer periphery of the dots. However, in the homonuclear (HO) QD molecule, the strain field was particularly intense along the connection zone that lies between the two dots. In this region, the presence of InP was visibly reduced, resulting in a lower overall height of the molecule, as depicted in Figure 17.

On the other hand, in the heteronuclear (HE) QD molecule, the strain field along the connection zone was comparatively less pronounced. Upon closer examination of the three-dimensional strain map, it becomes evident that the two dots in the HE QD molecule do not appear as separate and distinct structures, but rather as a single larger structure. This observation carries significant implications for the electronic coupling between the dots and signifies a qualitative difference between the HO and HE configurations.

## 10. Polytype GaAs/AlGaAs Quantum Dots

Polytypism represents a unique form of flexibility that enables precise manipulation of the electronic structure within a given material [88,89,90,91,92]. With advancements in nanowire growth techniques, such as atomically precise control of crystal-phase switching [93,94], it has become feasible to fabricate polytypic structures along the growth axis that are free from strain [95,96]. Remarkably, these structures can be scaled down to a size where quantum dots can be formed [97,98,99]. Although the wurtzite (Wz) phase is not naturally found in bulk A${}^{III}$B${}^{V}$ materials, except for nitrides, it can be achieved in nanowires, which has sparked significant interest in the scientific community due to its unique properties and technological implications [100,101,102]. Among these materials, Al*x*Ga1−xAs nanowires offer an intriguing platform for developing novel devices. By introducing the Al component to GaAs [103,104], the emission wavelength can be finely tuned over a wide range, enabling the fabrication of quantum devices [91,105].

As an illustrative case, let us consider an AlGaAs nanowire (NW) containing a GaAs quantum dot (QD). The AlGaAs nanowires were grown on a semi-insulating GaAs (111)B substrate using an EP1203 molecular beam epitaxy (MBE) system. The growth process involved the use of solid sources for Ga and Al atoms, as well as an As effusion cell to generate tetramers. To ensure the uniformity of the Al content, Raman spectroscopy was employed to monitor the Al concentration along the nanowire and within its cross-sections, thereby eliminating any potential inhomogeneities. The measured Al concentration ranged from c=0.24 to 0.26. TEM images of one of these AlGaAs NW clearly shows the evidence of phase transitions from Zincblende to Wurtzite crystal structures. Therefore, it was necessary to develop a model for electromechanical fields that takes into account this phase transition in the crystal.

Therefore, the continuum model utilized for the calculations of electromechanical fields in this study is based on the theory described in this paper. For clarity, in this section, we present again the equations required for the polytopic model. The change in total mechanical and electrical free energy density, denoted as dU, for a piezoelectric medium can be expressed as [29]:(98)$$\begin{array}{c} dU=dU^{mech}+dU^{elec}=TdS+\sigma_{ik}d\varepsilon_{ik}+E_{i}dD_{i}, \end{array}$$
where *T*, *S*, $\sigma_{ik}$, $\varepsilon_{ik}$, $E_{i}$, and $D_{i}$ are the temperature, entropy, stress tensor, strain tensor, electric field, and electric displacement, respectively. Under isentropic conditions (dS=0), and taking into account the crystal symmetry considerations, the elastic energy of a crystal with Zincblende (Zb) symmetry can be derived, as described in [25] as:(99)$$\begin{array}{cc} U_{Zb}^{mech} & =\frac{1}{2}[C_{11,Zb}\left(\varepsilon_{xx}^{2}+\varepsilon_{yy}^{2}+\varepsilon_{zz}^{2}\right)\\ \; & +2C_{12,Zb}\left(\varepsilon_{xx}\varepsilon_{yy}+\varepsilon_{xx}\varepsilon_{zz}+\varepsilon_{yy}\varepsilon_{zz}\right)\\ \; & +4C_{44,Zb}\left(\varepsilon_{xy}^{2}+\varepsilon_{xz}^{2}+\varepsilon_{yz}^{2}\right)], \end{array}$$
where $C_{ij,Zb}$ are the linearly independent stiffness constant parameters for a Zb structure.

Similarly, for a crystal with Wurtzite (Wz) symmetry we have for the elastic energy [25]:(100)$$\begin{array}{cc} U_{Wz}^{mech} & =\frac{1}{2}[C_{11,Wz}\left(\varepsilon_{xx}^{2}+\varepsilon_{yy}^{2}\right)+C_{33,Wz}\varepsilon_{zz}^{2}\\ \; & +2C_{12,Wz}\varepsilon_{xx}\varepsilon_{yy}+2C_{13,Wz}\varepsilon_{zz}\left(\varepsilon_{xx}+\varepsilon_{yy}\right)\\ \; & +4C_{44,Wz}\left(\varepsilon_{xz}^{2}+\varepsilon_{yz}^{2}\right)+2\left(C_{11,Wz}-C_{12,Wz}\right)\varepsilon_{xy}^{2}], \end{array}$$
where $C_{ij,Wz}$ are the linearly independent stiffness constant parameters for a Wz structure. Although the stiffness parameters for Zb GaAs, AlAs, and AlGaAs alloys can be found in the literature [106], the corresponding parameters for a Wz phase are not available. Therefore, we have estimated them using Martin’s transformations [107,108] based on the Zb parameters [109,110]:(101)$$\begin{array}{cc} C_{11,Wz} & =\frac{1}{6}\left(3C_{11,Zb}+3C_{12,Zb}+6C_{44,Zb}\right)\\ \; & -3\Delta^{2}/\left(C_{11,Zb}-C_{12,Zb}+C_{44,Zb}\right)\\ C_{12,Wz} & =\frac{1}{6}\left(C_{11,Zb}+5C_{12,Zb}-2C_{44,Zb}\right)\\ \; & +3\Delta^{2}/\left(C_{11,Zb}-C_{12,Zb}+C_{44,Zb}\right)\\ C_{13,Wz} & =\frac{1}{6}\left(2C_{11,Zb}+4C_{12,Zb}-4C_{44,Zb}\right)\\ C_{33,Wz} & =\frac{1}{6}\left(2C_{11,Zb}+4C_{12,Zb}+8C_{44,Zb}\right)\\ C_{44,Wz} & =\frac{1}{6}\left(2C_{11,Zb}-2C_{12,Zb}+2C_{44,Zb}\right)\\ \; & -6\Delta^{2}/\left(C_{11,Zb}-C_{12,Zb}+4C_{44,Zb}\right), \end{array}$$
where
(102)$$\begin{array}{c} \Delta \equiv \frac{\sqrt{2}}{6}\left(C_{11,Zb}-C_{12,Zb}-2C_{44,Zb}\right). \end{array}$$

The fully coupled continuum model we used incorporates the piezoelectric field directly [63]. As the Wz piezoelectric parameters are not available, we have approximated them using parameters from well-known Wz materials such as GaN, AlN, and AlGaN alloy. The governing equations for the electromechanical fields of the crystal were derived using expressions from (Equation 98) to (Equation 102) [63]. In the simulations, an Al${}_{c}$Ga${}_{1-c}$As nanowire (c=0.25) with a diameter of 40 nm and a height of 50 nm was considered. Within the nanowire, there was an embedded GaAs quantum dot with a diameter of 20 nm and a height of 5 nm. Two different cases were studied and compared: A pure Zincblende (Zb) case, where the entire structure was assumed to be grown in the pure Zb crystal phase, and a mixed structure. In the mixed structure, the GaAs quantum dot was sandwiched between two adjacent 8 nm high Wurtzite (Wz) layers and surrounded by a Zb AlGaAs shell. The mixed structure (Mx) was completed with Zb layers at the top and bottom. In Figure 19 the magnitude of the absolute value of the strain field for the pure Zb and the Mx structures is plotted.

In the mixed (Mx) case, in addition to the strain field typically observed inside and near the quantum dot due to the lattice mismatch between the materials, we also observed a weaker strain field at the interfaces between the two crystal phases of AlGaAs. Although this strain field was relatively weak, as expected due to the small lattice mismatch between GaAs and Al${}_{c}$Ga${}_{1-c}$As, as well as the even smaller crystal mismatch between Zincblende (Zb) and Wurtzite (Wz) AlGaAs, it was present throughout the nanowire. This strain field also has implications for the electrical properties of the structure.

Moreover, in the Mx case, the electric field resulting from the piezoelectric effect was relatively strong and exhibited a distinctive shape, which differed from the typical results obtained for Zincblende (Zb) quantum dots (QDs). This is evident in Figure 20, which displays the *z* component of the electric field for both Zb and Mx structures.

In contrast to the conventional quadrupole shape observed in Zincblende (Zb) quantum dots (QDs), the mixed (Mx) structure exhibited a Wurtzite (Wz)-like shape of the electric field in close proximity to and inside the dot. The electric field displayed typical dipole characteristics, with opposite signs inside and just outside the top and bottom of the dot. This unique behavior arose from the stronger piezoelectric effect induced by the Wz layers, despite the central volume of the column being in a pure Zb crystal phase. The enhanced magnitude of the piezoelectric effect in Wz heterostructures compared to Zb heterostructures has been previously reported in the literature [25].

## 11. Conclusions

In conclusion, this paper aimed to provide a comprehensive understanding of the mathematical models used to describe the electromechanical properties of heterostructure quantum dots. The focus was on both wurtzite and zincblende quantum dots, which have demonstrated significant relevance in optoelectronic applications.

The study presented a thorough overview of continuous and atomistic models for the electromechanical fields, covering a wide range of theoretical approaches. Notably, the paper introduced several unpublished approximations, including models based on cylindrical and cubic approximations. These last approaches allowed for the transformation of zincblende parametrization to wurtzite and vice versa, which is particularly useful in the case of polytypic quantum dots

Analytical results were supported by extensive numerical simulations, which encompassed a diverse set of scenarios. The numerical findings were also compared with experimental measurements, further validating the accuracy and applicability of the mathematical models.

Overall, this study contributes to the existing body of knowledge by offering a comprehensive and detailed understanding of the electromechanical properties of heterostructure quantum dots. The presented analytical models, along with the accompanying numerical results, provide valuable insights for the design and optimization of quantum dot-based optoelectronic devices.

## Figures and Tables

**Figure 1 nanomaterials-13-01820-f001:**
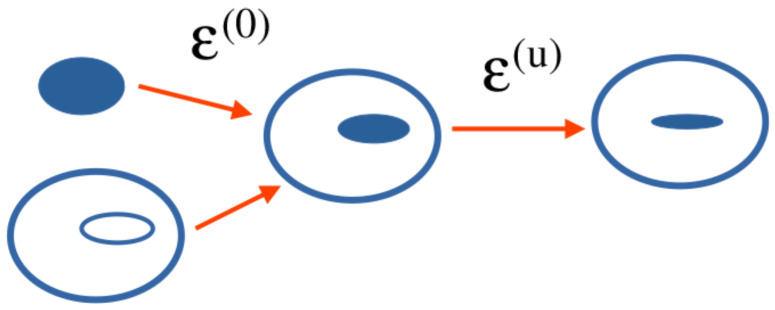
Tensor of local intrinsic strain and local strain tensor.

**Figure 2 nanomaterials-13-01820-f002:**
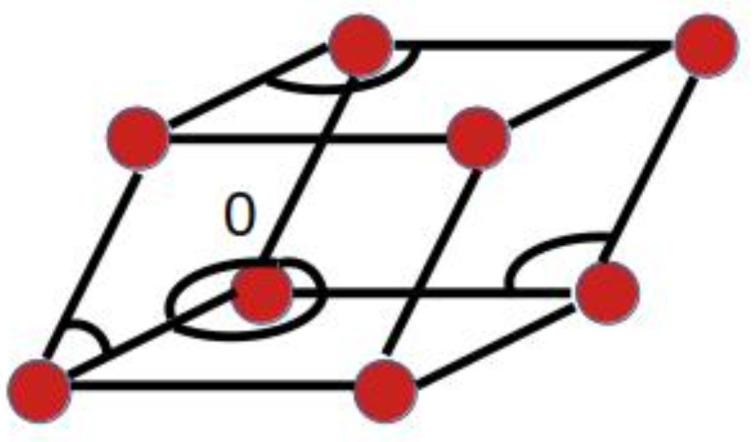
The unit cell with the six off-diagonal products.

**Figure 3 nanomaterials-13-01820-f003:**
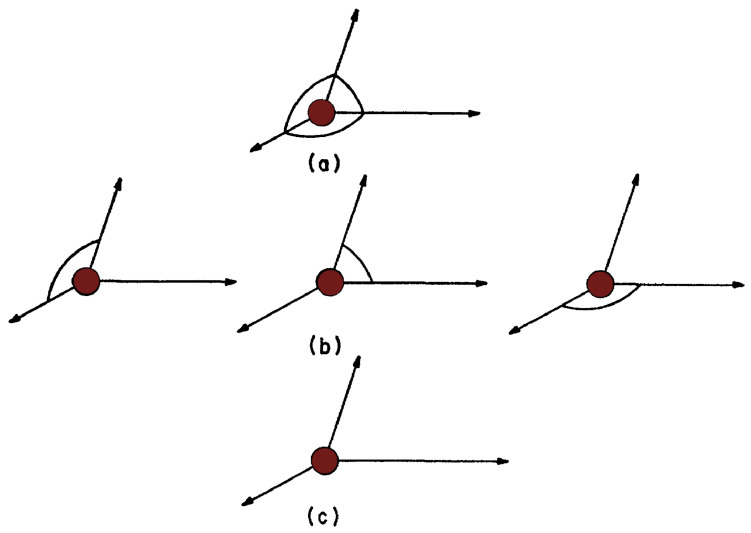
Types of lattice points according the number os associated scalar products: (**a**) points on the references lines, (**b**) points in the references planes but off the references lines, (**c**) ohter points.

**Figure 4 nanomaterials-13-01820-f004:**
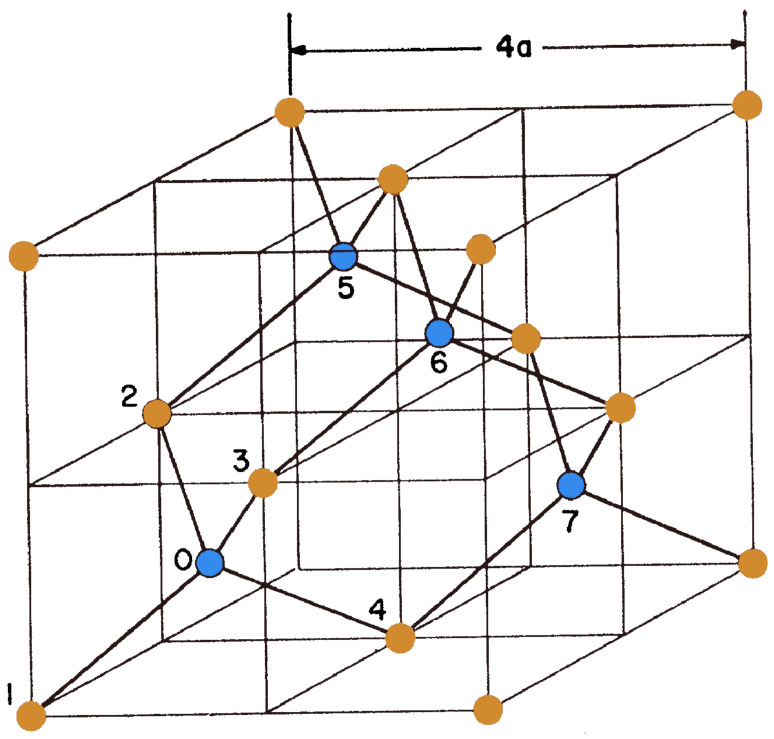
Crystal model for diamond structure. The atoms on the two different sublattice are represented by orange (atoms 1, 2, 3, 4) and blue (atoms 0, 5, 6, 7) circles.

**Figure 5 nanomaterials-13-01820-f005:**
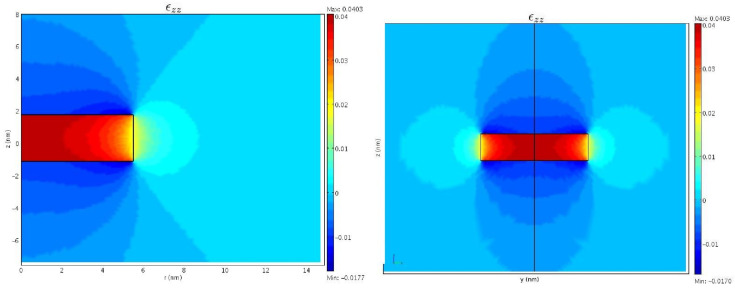
The strain tensor $\varepsilon_{zz}$ given by the two-dimensional model in the r,z plane (**left**) and by the three-dimensional model in the y,z plane (**right**).

**Figure 6 nanomaterials-13-01820-f006:**
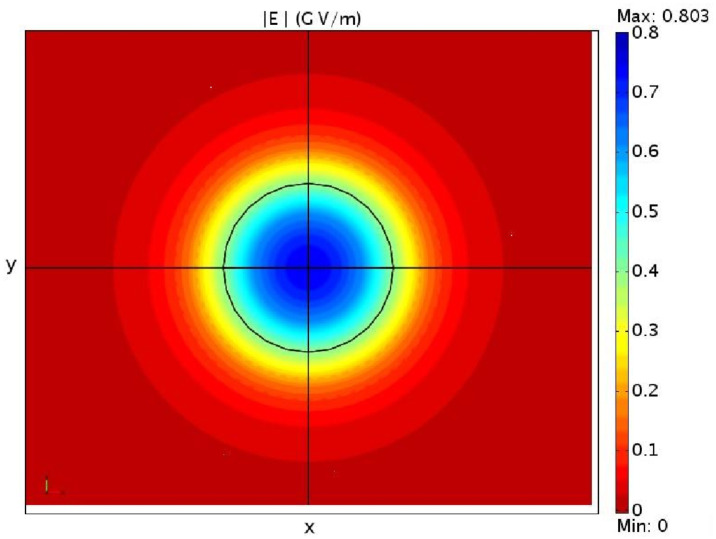
The absolute value of the electric field *E* in the x,y plane for a cylindrical GaN/AlN wurtzite quantum dot.

**Figure 7 nanomaterials-13-01820-f007:**
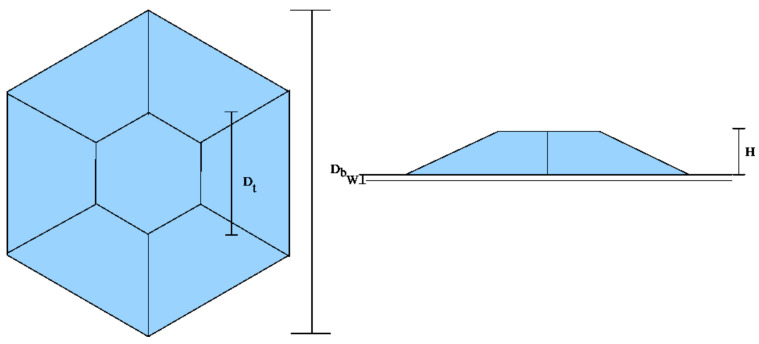
The geometry and the parameters of the hexagonal pyramid quantum dots.

**Figure 8 nanomaterials-13-01820-f008:**
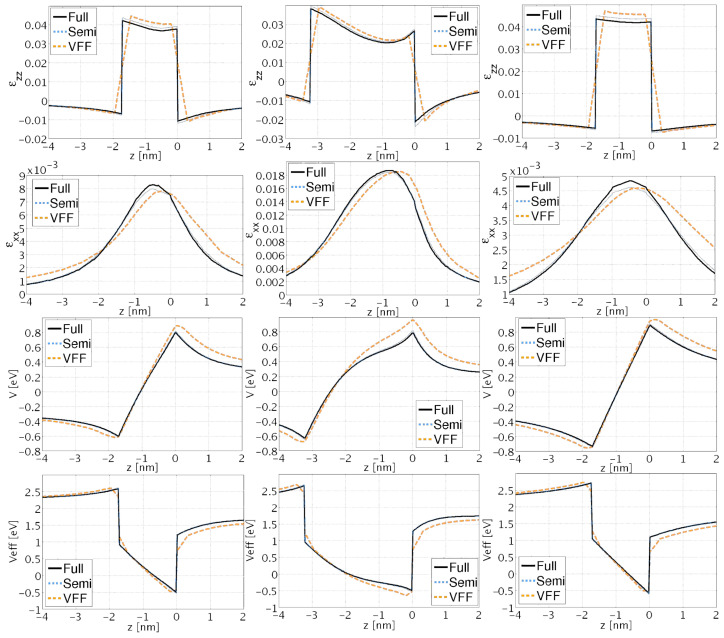
Solid black, dashed light blue, and dashed orange line codings correspond to fully-coupled continuum, semi-coupled continuum, and VFF data, respectively. The first, second, and third columns show results for Dot 1, 2, and 3, respectively.

**Figure 9 nanomaterials-13-01820-f009:**
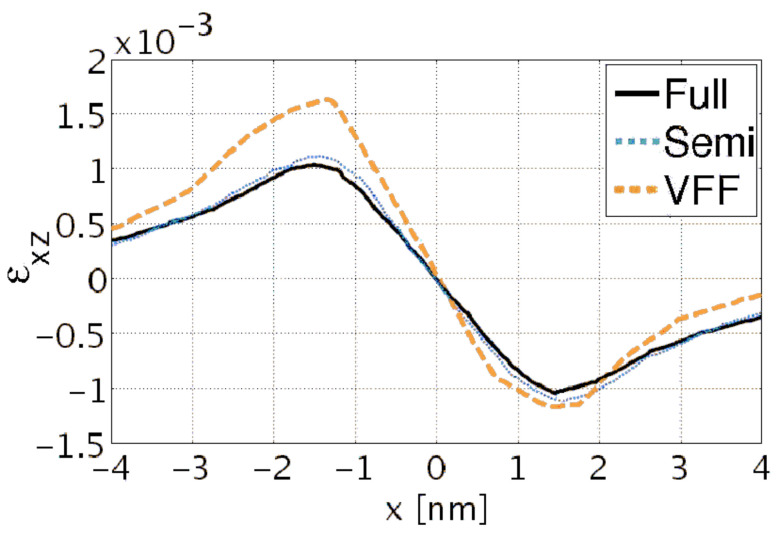
The $\varepsilon_{xz}$ strain component for Dot 1.

**Figure 10 nanomaterials-13-01820-f010:**
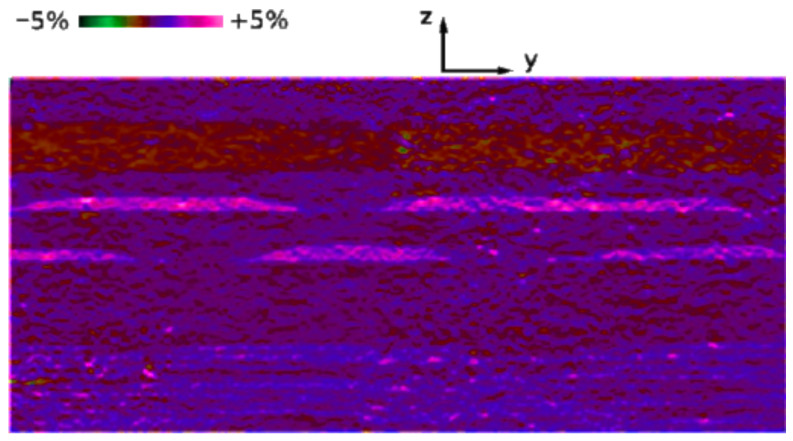
A 2D representation of the out-of-plane strain $\varepsilon_{zz}^{GaN}$, derived through geometric phase analysis of the HRTEM image of the sample, is shown. The image corresponds to the yz plane.

**Figure 11 nanomaterials-13-01820-f011:**
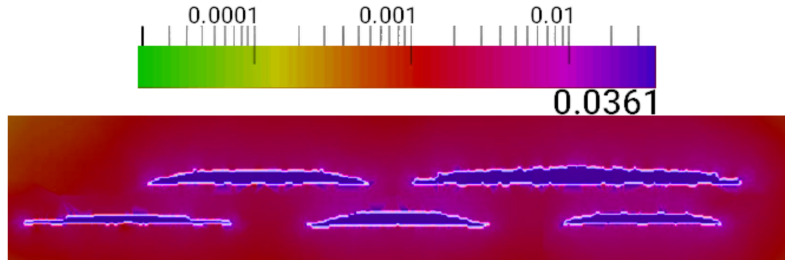
Plot of the hydrostatic strain magnitude in the active region, depicted in the yz plane, with the legend displayed in a logarithmic scale.

**Figure 12 nanomaterials-13-01820-f012:**
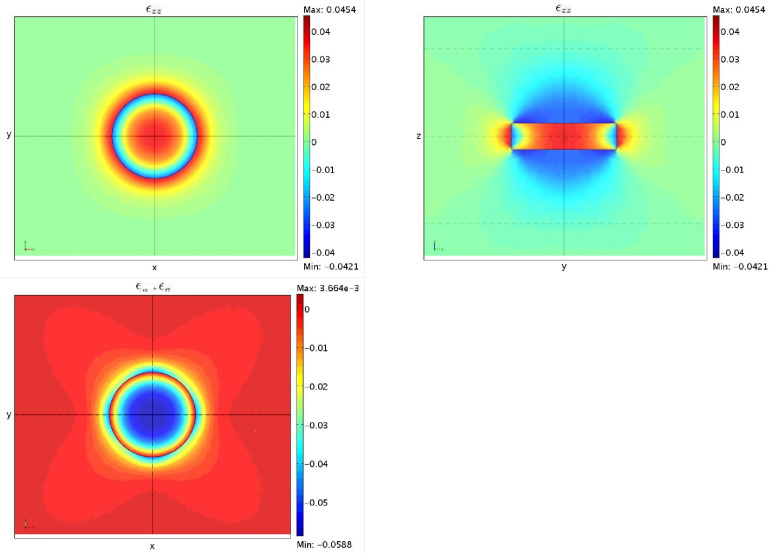
$\varepsilon_{zz}$ in the (x,y) (**top left**) and (y,z) (**top right**) plane, respectively, and $\left(\varepsilon_{xx}+\varepsilon_{yy}\right)$ in the (x,y) plane (**bottom**).

**Figure 13 nanomaterials-13-01820-f013:**
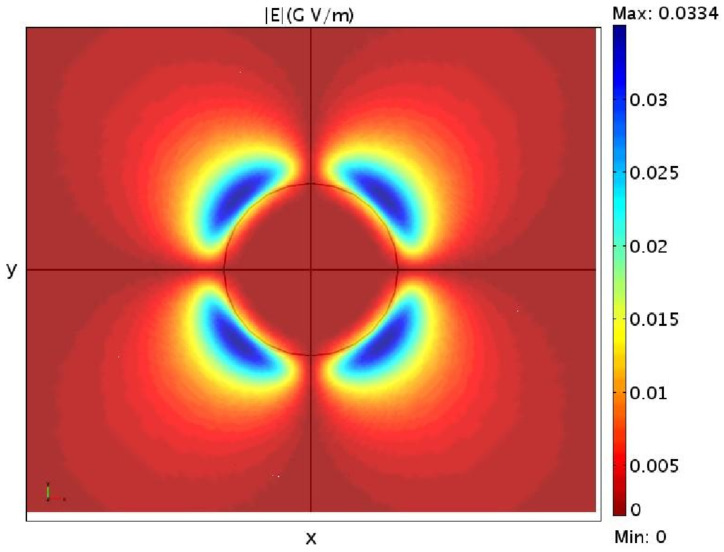
The absolute value of the electric field in the (x,y) plane.

**Figure 14 nanomaterials-13-01820-f014:**
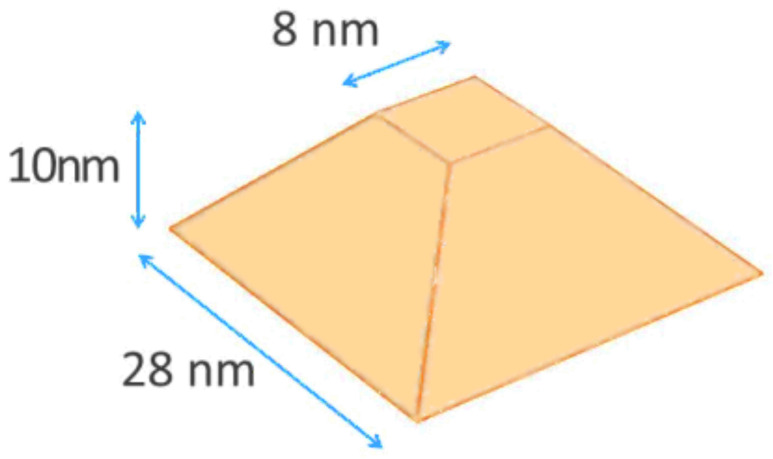
The geometry of the quantum dot under consideration. The *x*, *y*, and *z* directions are the [100], [010], and [001] directions, respectively.

**Figure 15 nanomaterials-13-01820-f015:**
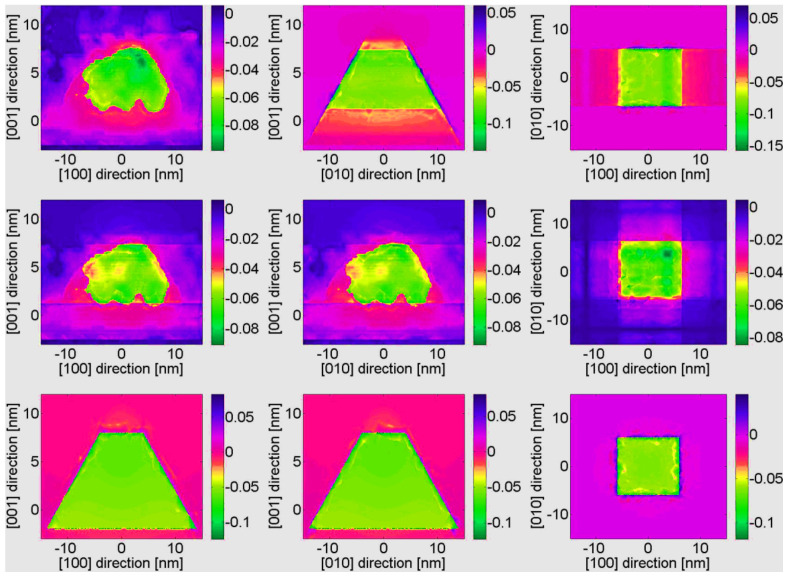
Hydrostatic strain component corresponding to indium concentration profile. Column 1, 2, and 3 represent the hydrostatic strain component for profiles 1, 2, and 3, respectively. The left plot illustrates the (010) plane, the middle plot depicts the (100) plane, both located at the dot’s center, and the right plot shows the (001) plane at a height of 6 nm.

**Figure 16 nanomaterials-13-01820-f016:**
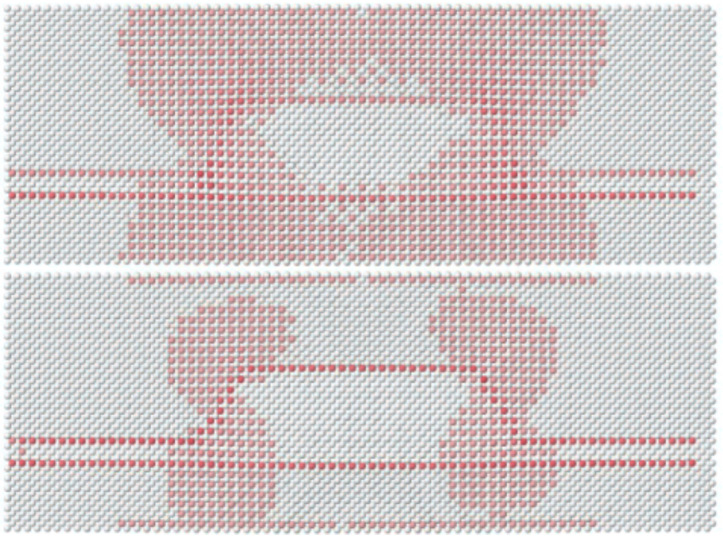
Top panel: standard deviation of the bond lengths after continuum model calculation on As atoms in xz plane. Bottom panel: the same after VFF. Color scale range between 0 (white) and 0.05 Å (red).

**Figure 17 nanomaterials-13-01820-f017:**
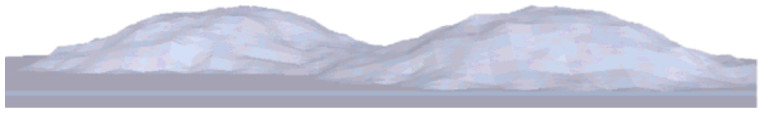
A 3D extrapolated structure of the homonuclear QD molecule.

**Figure 18 nanomaterials-13-01820-f018:**
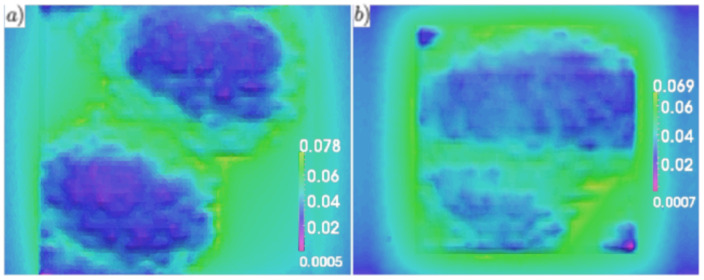
A top view of a three-dimensional plot depicting the magnitude of the hydrostatic strain field is shown for the homonuclear QD molecule (**a**) and the heteronuclear QD molecule (**b**).

**Figure 19 nanomaterials-13-01820-f019:**
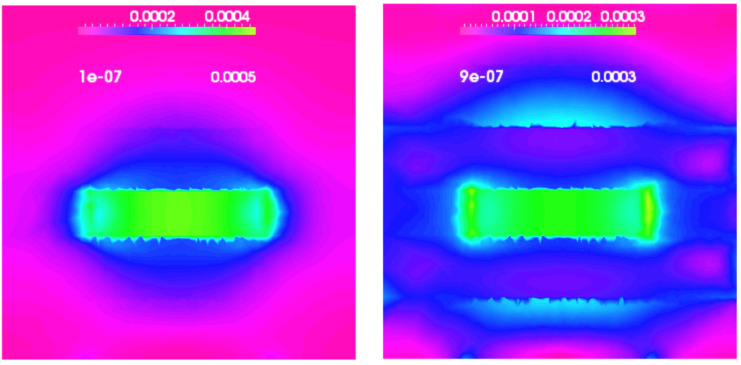
The magnitude of the absolute value of strain field for a pure Zb structure (**left**) and for the Mx structure (**right**).

**Figure 20 nanomaterials-13-01820-f020:**
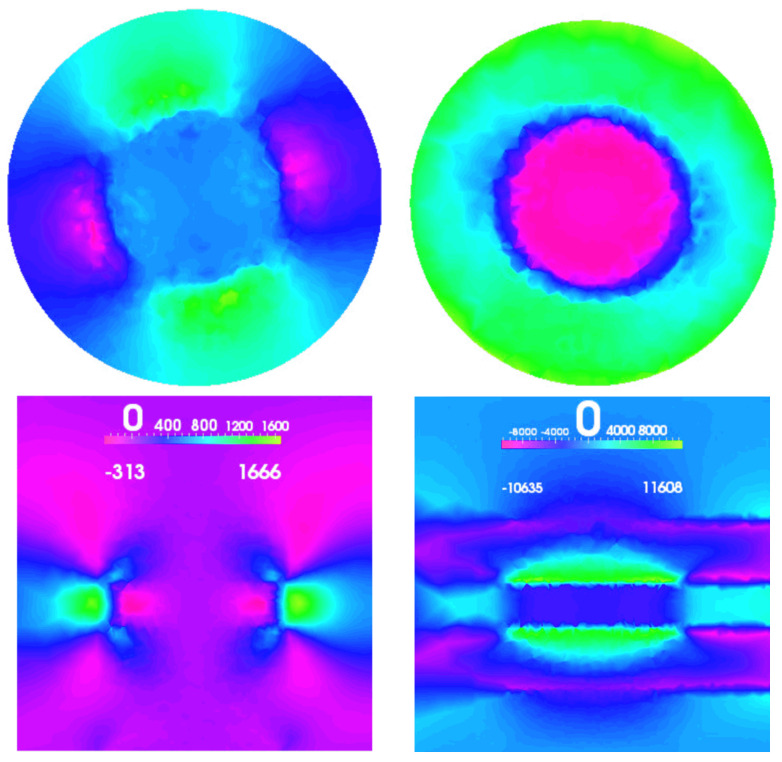
The *z* component of the electric field in the xy plane at the center of the quantum dot (**top**) and in the xz planes (**bottom**) is depicted. The left column corresponds to the Zincblende (Zb) structure, while the right column represents the mixed (Mx) structure. All values are given in GVolt/m.

**Table 1 nanomaterials-13-01820-t001:** The dimensions of the quantum dots.

	$D_{b}$ [nm]	$D_{t}$ [nm]	*H* [nm]	*W* [nm]
Dot 1	4.936	4.319	1.011	0.505
Dot 2	4.936	2.468	2.526	0.505
Dot 3	8.638	8.021	1.011	0.505

**Table 2 nanomaterials-13-01820-t002:** The groundstate energies of the quantum dots.

	Dot 1	Dot 2	Dot 3
Fully-coupled model (meV)	489	251	390
Semi-coupled model (meV)	474	237	371
Valence force field model (meV)	401	121	308

## Data Availability

The data presented in the current work are available on request from corresponding author.

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
