# Peer review of "State of the Art of Continuous and Atomistic Modeling of Electromechanical Properties of Semiconductor Quantum Dots"

_nanomaterials, 2023, doi:10.3390/nano13121820_

Round 1
Reviewer 1 Report
The author presents in this paper an exhaustive description of the mathematical models for the electromechanical properties of quantum dots. It is a well-organized and well written paper, but some improvements are still needed. Therefore, the author should address all the following questions before I can make the decision.
1. To show a bigger picture to the readers, the introduction about quantum dots could be expanded and more applications and advantages of QDs need to be discussed. This will let people understand why it is important to study QDs. The following recent example of QDs applications should be cited (https://doi.org/10.1039/C9PY01604J; https://doi.org/10.1021/acsmacrolett.9b00891).
2. Are figures 2 and 3 obtained from other papers? If so, please say copyright from what publishing group. Or the author can redraw the figure since it is not hard to draw.
3. Just curious, for the some recently discovered QD system, such as perovskite or graphene QDs, is there any model can applied to them?
Author Response
I thank the reviewers for their valuable comments that helped me in improving the manuscript. I respond to the comments in detail below.
Reviewer 1
The author presents in this paper an exhaustive description of the mathematical models for the electromechanical properties of quantum dots. It is a well-organized and well written paper, but some improvements are still needed. Therefore, the author should address all the following questions before I can make the decision.
I thank the reviewer for their extremely positive comment. Below are my responses:
1. To show a bigger picture to the readers, the introduction about quantum dots could be expanded and more applications and advantages of QDs need to be discussed. This will let people understand why it is important to study QDs. The following recent example of QDs applications should be cited (https://doi.org/10.1039/C9PY01604J; https://doi.org/10.1021/acsmacrolett.9b00891).
- I fully agree that there are countless applications for semiconductor quantum dots, and it is indeed challenging to mention them all. In response to the reviewer's suggestion, I have incorporated the following paragraph into the introduction:
There are numerous potential applications for semiconductor quantum dots. To highlight a few recent examples, we can consider Polymer-QDs nanocomposites, which are extensively studied and find utility in optical/electrical sensors, light-emitting diodes, as well as biological labeling/imaging [2]. Moreover, these nanocomposites hold promise as recyclable photocatalysts for aqueous PET-RAFT polymerization [3].
[2] Zhu, Y.; Egap E. PET-RAFT polymerization catalyzed by cadmium selenide quantum dots (QDs): Grafting-from QDs photocatalysts to make polymer nanocomposites. Polym. Chem., 2020,11, 1018-1024. https://doi.org/10.1039/C9PY01604J
[3] McClelland, K.P.; Clemons, T. D.; Stupp, S. I.; Weiss, E. A. Semiconductor Quantum Dots Are Efficient and Recyclable Photocatalysts for Aqueous PET-RAFT Polymerization. ACS Macro Lett. 2020, 9, 7-13. https://doi.org/10.1021/acsmacrolett.9b00891
2. Are figures 2 and 3 obtained from other papers? If so, please say copyright from what publishing group. Or the author can redraw the figure since it is not hard to draw.
- The two figures had nonetheless been taken similar to those in previous articles, but to avoid any misunderstandings, I followed the reviewer's advice and redid them from scratch.
3. Just curious, for the some recently discovered QD system, such as perovskite or graphene QDs, is there any model can applied to them?
- That is indeed a very valid curiosity. There are indeed studies on the possibility of using colloidal quantum dots in perovskite solar cells, as well as non-colloidal quantum dots. Additionally, there are numerous theoretical studies aiming to adapt models for strain and band structure calculations (such as k·p and tight binding) to devices incorporating quantum dots and two-dimensional materials, not limited to graphene. I am currently working on both of these scenarios and hope to publish something on this topic next year.
Reviewer 2 Report
The paper provides a comprehensive overview of the mathematical models used to describe the electromechanical properties of heterostructure quantum dots, with a focus on both wurtzite and zincblende quantum dots. The paper discusses the challenges in modeling these properties and presents key findings from recent research. The potential applications of this research are also discussed.
The paper in the current form requires major edition to address the following issues:
1) There are some definitions that are standard to the field and vary basic, such as stress and strain that can be found in any textbook. The authors shall move those sections to appendix or suitable them with reference to some text book. would be more suitable.
2) There are several paragraphs that are very short, and even 1 sentence. A paragraph needs to be at least 4-5 sentences.
3) It is not clear how the authors got to eq 12 from eq 11. Why the sign of dW changed in Eq 12?
4) Several assumptions are made that are not clear how they are derived or applicable to the case of quantum dots, these include (assuming constant Temp, then isotropic material, and no external force).
5) Why Helmholtz free energy was chosen rather than Gibbs?
6) there is a type in Helmholtz
7) Eq (28) is not correct. The total strain calculated by eq (5) is the epsilon not epsilon^u. epsilon^u is the elastic strain and not total strain. Subsequently Fig 1 is not appropriate.
8) There are several refs on this topic that are missing, including ones that address the solution of the equations in cylindrical coordinate system that intrinsically is singular along the axis of symmetry. e.g.,
Nano Lett, 11 (2011), p. 786
Acta materialia 60 (13-14), 5117-5124
Nano Lett, 9 (2009), p. 4177
Journal of Applied Physics 108 (11), 114303
Int J Numer Meth Eng, 84 (2010), p. 1541
Phys. Chem. Chem. Phys. 16 (10), 4522-4527
The English language requires major editing. There are several one sentence paragraphs that need to be combined with other paragraphs to make the paper readable.
Author Response
I thank the reviewers for their valuable comments that helped me in improving the manuscript. I respond to the comments in detail below.
The paper provides a comprehensive overview of the mathematical models used to describe the electromechanical properties of heterostructure quantum dots, with a focus on both wurtzite and zincblende quantum dots. The paper discusses the challenges in modeling these properties and presents key findings from recent research. The potential applications of this research are also discussed.
The paper in the current form requires major edition to address the following issues:
1) There are some definitions that are standard to the field and vary basic, such as stress and strain that can be found in any textbook. The authors shall move those sections to appendix or suitable them with reference to some text book. would be more suitable.
- I understand the reviewer's objection, but the underlying idea was precisely to start from the basic equations of crystal mechanics and develop a theory applicable to a wide range of quantum dot scenarios. The theoretical part was designed to be a didactic step-by-step treatment, accessible even to students or doctoral candidates in the field, also, precisely in order to unify all the material found in various texts.
2) There are several paragraphs that are very short, and even 1 sentence. A paragraph needs to be at least 4-5 sentences.
- I thank the reviewer for pointing out this particular detail, and I have made the necessary modifications to the text accordingly.
3) It is not clear how the authors got to eq 12 from eq 11. Why the sign of dW changed in Eq 12?
- I thank the reviewer for pointing out the typo, which I have corrected.
4) Several assumptions are made that are not clear how they are derived or applicable to the case of quantum dots, these include (assuming constant Temp, then isotropic material, and no external force).
- The mathematical assumptions are made for the sack of simplicity, but in reality, there are no specific limitations in using, for example, the continuous model beyond them, such as in the presence of an external force (see for example D. Barettin et al J. Phys. Mater. 4 (2021) 034008) or non-homogeneous structures (see ad example D. Barettin et al. Nanomaterials 2023, 13, 1367.), or even in other more complex cases.
5) Why Helmholtz free energy was chosen rather than Gibbs?
- In the thermodynamic treatment, I faithfully followed the text of Landau-Lifshitz, see Ref. [31].
6) there is a type in Helmholtz.
Thank you for pointing it, I have corrected it.
7) Eq (28) is not correct. The total strain calculated by eq (5) is the epsilon not epsilon^u. epsilon^u is the elastic strain and not total strain. Subsequently Fig 1 is not appropriate.
- For equation (28), there are two possible choices arising from two different definitions of the lattice mismatch: one from the work of Chuang and Chang (see Ref. [26]) and one from the work of Fonoborov-Balandin (see Ref. [25]). In this case, I have chosen the first one, which is also used in Ref[43] as in the present paper.
8) There are several refs on this topic that are missing, including ones that address the solution of the equations in cylindrical coordinate system that intrinsically is singular along the axis of symmetry. e.g.,
Nano Lett, 11 (2011), p. 786
Acta materialia 60 (13-14), 5117-5124
Nano Lett, 9 (2009), p. 4177
Journal of Applied Physics 108 (11), 114303
Int J Numer Meth Eng, 84 (2010), p. 1541
Phys. Chem. Chem. Phys. 16 (10), 4522-4527
- Certainly, the topic was extremely broad, so I had to make a selection of the topics to present. I thank the reviewer for bringing these publications to my attention. Therefore, I have added the following sentence in the section dedicated to the cylindrical approximation:
"In some configurations, it is possible to solve the electromechanical problem using a cylindrical approximation. For a more detailed discussion on different aspects related to the cylindrical approximation, it is possible refer to Ref [28-33]
Reviewer 3 Report
This is a review paper. The author spent a significant portion of the paper on well-known knowledge, which can be found in textbooks, and provided some examples.
In general, this is an interesting review. The following is a list of concerns.
1. Please condense the equations for continuum mechanics. The author can refer readers to related textbooks.
2. Please compare Eq. 49 with Eq. 84 and make them consistent.
3. Please discuss the effects of interface on the EM interaction.
4. For QDs, surface effects can play an important role in EM effects. Please discuss such effects.
Author Response
I thank the reviewers for their valuable comments that helped me in improving the manuscript. I respond to the comments in detail below.
This is a review paper. The author spent a significant portion of the paper on well-known knowledge, which can be found in textbooks, and provided some examples.
In general, this is an interesting review. The following is a list of concerns.
1. Please condense the equations for continuum mechanics. The author can refer readers to related textbooks.
-- I understand the reviewer's objection, but the underlying idea was precisely to start from the basic equations of crystal mechanics and develop a theory applicable to a wide range of quantum dot scenarios. The theoretical part was designed to be a didactic step-by-step treatment, accessible even to students or doctoral candidates in the field, also, precisely in order to unify all the material found in various texts.
2. Please compare Eq. 49 with Eq. 84 and make them consistent.
- I really thank the reviewer for noticing this inconsistency, the equations are now consistent.
3. Please discuss the effects of interface on the EM interaction.
4. For QDs, surface effects can play an important role in EM effects. Please discuss such effects.
- I thank the reviewer for pointing out these two aspects. I have included the following sentence in Section 4:
Traditional elasticity does not exhibit inherent size dependence in the elastic solutions of embedded inhomogeneities. In systems with dimensions larger than 50 nm, the surface-to-volume ratio is usually insignificant, and the deformation behavior is primarily governed by classical bulk strain energy. Presently, there is no available framework that combines interface and surface elasticity with bulk elasticity to analyze embedded inclusions. The approach utilized in quantum dot literature relies solely on classical bulk elasticity. For more information on corrections related to hydrostatic strain arising from interfacial and surface elasticity, please refer to Ref [48, 49].
Round 2
Reviewer 2 Report
total strain is the symmetric part of the deformation gradient tensor, and is the sum of elastic and intrinsic (transformation) terms, which needs to be corrected. see eq 28
The ref list need to be corrected e.g., ref 25 doi is out of the page.
Author Response
I thank the reviewer for the additional comments.
The following sentence has been added regarding equation (28):
"the contribution of internal strain has been neglected in equation (28)."
The reference has been corrected as indicated.
Reviewer 3 Report
Can be published.
Author Response
I thank the reviewer for his valuable comments.